# Discovering Failure Modes of Text-guided Diffusion Models via Adversarial Search

**Qihao Liu**[1]    **Adam Kortylewski**[2,3]    **Yutong Bai**[1]    **Song Bai**[4]    **Alan Yuille**[1]
[1]Johns Hopkins University    [2]University of Freiburg
[3]Max-Planck-Institute for Informatics    [4]ByteDance Inc.

## Abstract

Text-guided diffusion models (TDMs) are widely applied but can fail unexpectedly. Common failures include: *(i)* natural-looking text prompts generating images with the wrong content, or *(ii)* different random samples of the latent variables that generate vastly different, and even unrelated, outputs despite being conditioned on the same text prompt. In this work, we aim to study and understand the failure modes of TDMs in more detail. To achieve this, we propose SAGE, the first adversarial search method on TDMs that systematically explores the discrete prompt space and the high-dimensional latent space, to automatically discover undesirable behaviors and failure cases in image generation. We use image classifiers as surrogate loss functions during searching, and employ human inspections to validate the identified failures. For the first time, our method enables efficient exploration of both the discrete and intricate human language space and the challenging latent space, overcoming the gradient vanishing problem. Then, we demonstrate the effectiveness of SAGE on five widely used generative models and reveal four typical failure modes that have not been systematically studied before: (1) We find a variety of natural text prompts that generate images failing to capture the semantics of input texts. We further discuss the underlying causes and potential solutions based on the results. (2) We find regions in the latent space that lead to distorted images independent of the text prompt, suggesting that parts of the latent space are not well-structured. (3) We also find latent samples that result in natural-looking images unrelated to the text prompt, implying a possible misalignment between the latent and prompt spaces. (4) By appending a single adversarial token embedding to any input prompts, we can generate a variety of specified target objects, with minimal impact on CLIP scores, demonstrating the fragility of language representations. Project page: https://sage-diffusion.github.io

## 1 Introduction

Text-guided Diffusion Models (TDMs) (Rombach et al., 2022; Saharia et al., 2022) are highly expressive models that have recently achieved state-of-the-art performance in image generation tasks. However, TDMs are black-box models that can exhibit undesirable behaviors, making it of great importance to understand when and how they fail. Therefore, immediately after the introduction of DALL·E-2 (Ramesh et al., 2022) in 2022, a considerable research interest evolved around discovering and understanding the failure modes of TDMs (Marcus et al., 2022; Maus et al., 2023; Wen et al., 2023; Zhuang et al., 2023; Leivada et al., 2022; Gokhale et al., 2022; Conwell & Ullman, 2023).

However, automatically detecting these behaviors in generative models is challenging. Previous failure detection methods could only focus on the language part of TDMs, but neglected the latent space due to its high-dimensional complexity and not being easily differentiable. In addition, even for the language space, due to the discrete nature and the intricate patterns of human languages, previous methods have to either rely on the searching of the CLIP embedding space (Radford et al., 2021) and generating non-interpretable gibberish, or manually sample different input prompts to test TDMs, which is time-consuming and inefficient. More importantly, this manual approach heavily constrains the failure analysis process, causing previous methods to mostly focus on evident failures such as compositionality, but researchers nevertheless have tried to improve TDMs based on this identified failure (Liu et al., 2022; Kumari et al., 2023; Chefer et al., 2023).

In this work, we propose the first failure search method that aims to automatically and systematically discover the failure modes of TDMs by exploring *all three input spaces*: the latent space, token space,

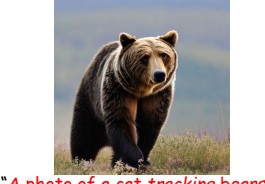 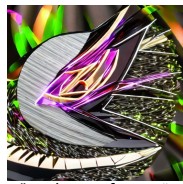 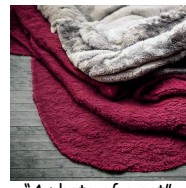 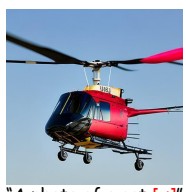

"A photo of a cat _tracking_ bears"
*Z*: Randomly sampled
Failure generation rate: 96%
(a)

"A photo of a cat"
*Z*: Optimized (type-1)
Failure generation rate: 100%
(b)

"A photo of a cat"
*Z*: Optimized (type-2)
Failure generation rate: 100%
(c)

"A photo of a cat [×]"
*Z*: Randomly sampled
Failure generation rate: 92%
(d)

Figure 1: We propose the first adversarial search method to automatically, effectively, and systematically find failure cases of any text-guided diffusion models in the human-understandable language space, latent space ($z$), and token space. Our proposed method identifies four typical failure modes of TDMs as highlighted in red: (a) We find a variety of natural text prompts that TDMs find unintelligible, including previously unexplored failures such as the omission of objects due to specific actions (*e.g.* "tracking"). We report two types of errors in the latent space that lead to either (b) distorted images (type-1) or (c) natural-looking but unrelated images (type-2). (d) We find token embeddings that overwrite different input prompts and generate images with only the specified target objects.

and human-understandable language space. To achieve this, we propose an adversarial optimization approach, which takes inspiration from adversarial attacks on discriminative models (Szegedy et al., 2013), but resolves several open challenges that ultimately enable us to discover failure modes in TDMs. In particular, we make three methodological contributions:

1. Adversarial attacks assume an objective that can be optimized. We propose to use the output of image classifiers as a **surrogate loss function**. In practice, however, we observe that images with high surrogate losses are mostly caused by classifier errors in detecting target objects, rather than the failure of the diffusion model in generating them. To tackle this, we design an ensemble of robust classifiers to enhance the effectiveness of the surrogate loss in detecting true failures of TDMs. Human studies show that when attacking our ensemble surrogate loss, over eighty percent of the failures are due to diffusion models.

2. A simple gradient-based optimization over the latent space in TDMs is not feasible, since they typically require hundreds of sampling steps (Ho et al., 2020) through a U-Net (Ronneberger et al., 2015), which causes vanishing gradients when tracing back from the output to the latent space. We show that this can be addressed, by **adding residual connection and using approximate gradients** computed from the earlier layers.

3. Optimizing over the natural language space poses an additional challenge since human language follows complex patterns, and the discrete nature of the text makes it hard to optimize. We address these issues by using LLMs to provide language priors, and **introducing a gradient-guided optimization process** that first optimizes a continuous token embedding and subsequently uses the continuous gradients to guide the discrete search in the prompt space.

We refer to our system as SAGE, which stands for **S**earching **A**dversarial Failures in Text-guided **Ge**nerative Models. Our experiments show that SAGE can effectively identify a variety of failure cases of different TDMs and GAN-based models. Then, with SAGE, we give a comprehensive analysis of SOTA generative models, and investigate in detail **four typical failure modes across all TDMs and GAN-based models**, including GLIDE (Nichol et al., 2021), Stable Diffusion V1.5/V2.1 (Rombach et al., 2022), DeepFloyd (Konstantinov et al., 2023), and StyleGAN-XL (Sauer et al., 2022):

Firstly, we find **natural text prompts that are unintelligible for diffusion models (Fig. 1 a)**. Beyond the most evident failure reported previously (*i.e.*, compositionality), we also discover and investigate several previously-unexplored interesting phenomena, such as the failures due to specific actions (*e.g.*, in Fig. 1 a, the word "tracking" causes 96% of the images to miss the cat), adjectives, salient features in words, and so on. Secondly, we find **connected regions in the latent space that consistently lead to distorted images (Fig. 1 b)**. It indicates irregularities in certain regions of the latent space, prompting the need for further study to enhance TDMs. Thirdly, we find **latent samples that do not depict the key objects but associated backgrounds (Fig. 1 c)**, suggesting that the latent space and prompt space are not fully aligned. We also show that this failure is directly related to the model stability of TDMs, as defined in Sec. 5.3. Finally, as a curiosity, we also find **token embeddings that overwrite the input prompts (Fig. 1 d)**, causing the diffusion model to produce irrelevant images with only the specified target object across a variety of input prompts. It suggests that the current language representations are sensitive and biased.

In summary, we make the following contributions:

- We proposed SAGE, the first automated method to systematically find failure modes of any TDMs and other generative models. To the best of our knowledge, we are the first to successfully attack the latent space and the first to automatically find interpretable prompts in real language space.
- Using SAGE, we conduct a comprehensive analysis of the failure modes of SOTA generative models. Our investigation delves deeply into several previously unexplored phenomena. We then analyze the underlying causes of these failures and potential solutions to improve the robustness.

## 2    RELATED WORK

**Diffusion models.** Diffusion models have shown impressive accomplishments in image generations (Rombach et al., 2022; Nichol et al., 2021; Ho et al., 2020). Numerous studies have focused on enhancing them by scaling them up or accelerating the sampling procedure (Dhariwal et al., 2021; Ramesh et al., 2022; Saharia et al., 2022; Rombach et al., 2022; Ho et al., 2020; Kong & Ping, 2021; Lu et al., 2022). They're applied in diverse domains like music/video/3D generation and reinforcement learning (Yang et al., 2022; Wang et al., 2022; Poole et al., 2022; Huang et al., 2023).

**Prompt perturbations in diffusion models.** Recent work (Daras & Dimakis, 2022; Maus et al., 2023; Wen et al., 2023; Millière, 2022; Zhuang et al., 2023) studies the sensitivity of TDMs to prompt perturbations. However, their approach is restricted to detecting gibberish that is not interpretable. Another line of work (Marcus et al., 2022; Leivada et al., 2022; Gokhale et al., 2022; Conwell & Ullman, 2023) studies the languages that cannot be understood by TDMs. However, they still need to manually generate those prompts, while our method allows for an automatic and systematic exploration of the entire language space to find model weaknesses. In addition, they only focused on perturbations in prompt space, ignoring the sensitivity in latent space. Nonetheless, our study shows that the latent space also has many intriguing properties that are important and worth studying.

**Adversarial attacks.** SAGE adopt the concept of adversarial attacks (Goodfellow et al., 2014; Szegedy et al., 2013; Madry et al., 2017), which deceive neural networks by integrating tiny perturbations into inputs. This is conceptually similar to our failure search in the latent space. However, attacking the latent space of large diffusion models with standard adversarial attack is actually challenging due to GPU limitations and gradient vanishing issues. And even if we resolve these obstacles with engineering efforts such as FP16 and parameter freezing during gradient computation, and run it on A100 GPUs with 80G RAM, the success rate remains remarkably low compared to our method. In contrast, SAGE enables an efficient and low-cost search/optimization in the latent space of TDMs.

**Failure cases of diffusion models.** One prominent failure mode that was observed in prior work is the compositionality (Gokhale et al., 2022; Marcus et al., 2022; Conwell & Ullman, 2023). To address this issue, one line of work factorizes the generation of multiple objects by explicitly composing different diffusion models (Liu et al., 2022; Du et al., 2023; Kumari et al., 2023; Bar-Tal et al., 2023), while alternative methods use guidance from attention and/or network activation of diffusion models to compose multiple objects into one image (Epstein et al., 2023; Chefer et al., 2023; Feng et al., 2022). These publications underscore the importance of identifying failure modes in diffusion models. However, until now, the discovery of failure modes has largely relied on manually crafted prompts, imposing significant limitations on the failure analysis process. As a result, prior work mainly focuses on resolving problems related to compositionality. Our automated failure discovery, on the other hand, reveals that many other failure modes exist due to, for example, adjectives, specific motions, or salient features in words, which are equally important and need to be addressed.

## 3    METHODOLOGY

Figure 2 shows an overview of our pipeline. We study text-guided diffusion model $\mathcal{G}$, that takes as input text prompt $p$ and latent variable $z \sim \mathcal{N}(\mathbf{0}, \mathbf{I})$ to generate an image $I = \mathcal{G}(p, z)$. Our goal is to search over the latent space and the text space of model $\mathcal{G}$ to detect failure modes. In order to measure if the model output is valid or not, we constrain the generative model using a fixed template for the first words of the input prompt, i.e. "A photo of a [class]", where [class] describes a key object class. This enables us to use an image discriminator $\mathcal{D}(\mathcal{I})$ to measure if the key [class] is present in the generated image $I$. Building on this pipeline, we can formulate an adversarial optimization process over the prompt $p$ and the latent code $z$ to search for images that do not depict the key object. However, this optimization process requires resolving several open challenges:

*Firstly*, adversarial attacks assume an objective function. We discuss in Sec. 3.4 how to design a robust discriminator to guide the search towards finding true failures of diffusion models. *Secondly*, The

Figure 2: **Overview.** Given a text-guided generative model $\mathcal{G}$, we want to automatically find natural text prompts $p$ or non-outlier latent variables $z$ that generate failure images. We formulate it as an adversarial optimization process. A gradient-guided search policy $\mathcal{P}$ is proposed to enable efficient search over the discrete prompt space, and the residual connection is used to back-propagate the gradient from outputs $I$ to the inputs of latent variable. Finally, an ensemble of discriminative models is used to obtain a robust discriminator $\mathcal{D}$ to find true failures of the generative model $\mathcal{G}$.

text space is discrete, and thus cannot be easily optimized using gradient-based methods. To resolve this problem, our approach first starts by optimizing a continuous token embedding (Sec. 3.1) and subsequently uses the continuous gradients to guide the discrete search in the prompt space (Sec. 3.2). *Finally*, Computing the gradient of the latent code $z$ w.r.t. to the model output is challenging because of the iterative image generation process. In Sec. 3.3, we discuss how this problem can be overcome, by using an approximate gradient from the intermediate layers of the diffusion model.

## 3.1 ADVERSARIAL OPTIMIZATION OF TOKEN EMBEDDINGS

TDMs typically use a pre-trained language model, such as CLIP, as an encoder to process the input text prompt $p$ into a token embedding $\tau = \tau_\theta(p)$, which is subsequently used to condition the image generation process. To initialize the optimization process, we append a single token to the template prompt "A photo of a [class] [x]", where [class] is the key object, and [x] is the token we are optimizing in an adversarial manner. We initialize the embedding $\tau$ of [x] with zero. Unless otherwise specified, $\tau_{[x]}$ has the dimensionality of a single token.

As the embedding $\tau$ conditions the generation process at every diffusion step, we can directly back-propagate gradients from the outputs to $\tau$. Projected gradient descent (Madry et al., 2017) is adopted for optimization. In practice, we found that perturbing the gradients stochastically leads to better results as it prevents model from getting stuck in the local optimum. Hence, we optimize $\tau$ following:

$$\tau_{[x]} = \tau_{[x]} + r \cdot \alpha \cdot \text{sgn}(\nabla_{\tau_{[x]}} \mathcal{L}_c(z, \tau)), \tag{1}$$

where $\mathcal{L}_c(z, \tau) = -\mathcal{D}(\mathcal{G}(z, \tau))$, $\mathcal{G}$ is the generator and $\mathcal{D}$ is the robust discriminator detailed in Sec. 3.4, $z$ is a latent code that is fixed during optimization, $\alpha$ is the step size, and $r \in [0, 1]$ is a random value. We also constrain the optimized embedding to be in the range $\tau_{[x]} \in [-2.5, 2.5]$.

## 3.2 GRADIENT-GUIDED SEARCH FOR TEXT PROMPTS

The gradient-based optimization of token embeddings (Sec 3.1) enables us to find adversarial examples in the embedding space, but these are usually far from the word embeddings of human language. Therefore, we further constrain the optimization process by introducing a language prior that is provided by a pretrained language model (*e.g.*, LLaMA (Touvron et al., 2023)).

Intuitively, we want the optimized embedding to be adversarial, while at the same time being close to the embeddings of real words. However, we cannot simply use the embeddings of all words that exist in language, as this would be computationally infeasible. Instead, we use the pretrained language model to generate a list of word proposals. Specifically, we first initialize a prompt with "A photo of a [class]", where [class] is the key object. Given this template, we use the language model to generate $k$ possible words (*i.e.* $k$ candidates [c]) that can be added after the input, then we compute the token embeddings $\tau_{[c]}$ of all candidates and use them as the constraints during optimization.

In practice, we found that instead of directly optimizing for the next word in the text prompt (denoted by [x]), it is more effective to also take into account the subsequent embedding after attaching [x] to the input prompt (*i.e.*, to also take into account the expected loss in future embeddings). Intuitively, while [x] itself may not directly lead to model failures, the embedding space after adding [x] to the input prompt may be much easier to attack. Therefore, we consider two words [x, y] during the optimization, where [x] remains the search target and [y] is the word after [x].

We optimize the embedding by using the proposal embeddings $\tau_{[c]}$ as additional constraints to (1) initialize the target embedding $\tau_{[x]}$ by randomly sampling the embedding within the range of values seen in $\tau_{[c]}$, and (2) add an additional optimization constraint such that the optimized embedding has

a small distance to the corresponding nearest candidate $[\mathtt{c_n}]$ in the embedding space. Specifically , we perform updates following:

$$\tau_{[\mathtt{x,y}]} = \tau_{[\mathtt{x,y}]} + r \cdot \alpha \cdot \mathrm{sgn}(\nabla_{\tau_{[\mathtt{x,y}]}} \mathcal{L}_d(z, \tau)) \tag{2}$$

$$\mathcal{L}_d(z, \tau) = -\mathcal{D}(\mathcal{G}(z, \tau)) + \lambda \mathcal{S}(\tau_{[\mathtt{x}]}, \tau_{[\mathtt{c_n}]}) \tag{3}$$

where $\lambda$ is a scalar weight and $\mathcal{S}(\cdot)$ computes the cosine similarity between two embeddings.

After convergence, we compute the similarity again and choose the closest candidate $[\mathtt{w}] = \arg\min_n \mathcal{S}(\tau_{[\mathtt{x}]}, \tau_{[\mathtt{c_n}]})$ as the next word to be added to the current input prompt (*i.e.* "A photo of a $[\mathtt{class}]\ [\mathtt{w}]$"). We repeat this process for a maximum of $m = 10$ times or until a sentence is complete. We employ LLaMA here, but any standard text generator can be used. We include an ablation about different text generators in Supp. B.8, and the use of template prompt in Supp. B.9.

## 3.3 ATTACKING THE LATENT SPACE

Different from Sec 3.1 where we directly search for the embedding $\tau_{[\mathtt{x}]}$, our goal when attacking the latent space is to perturb a given latent code $z$ minimally, similar to standard attack problems. Specifically, we aim to find a small perturbation $d_z$ such that for a randomly sampled input latent code $z_T$, $z_T + d_z$ leads to a failure in the generation process ($z_T$ represents the same thing as $z$ in other sections of this paper). We initialize $d_z$ with zero and update it using the surrogate loss:

$$d_z = d_z + r \cdot \alpha \cdot \mathrm{sgn}(\nabla_{d_z} \mathcal{L}_c(z_T + d_z, \tau)) \tag{4}$$

where $\mathcal{L}_c(z_T + d_z, \tau)) = -\mathcal{D}(\mathcal{G}(z_T + d_z, \tau))$. We constrain the perturbation to be in the range $d_z \in [-1, 1]$. However, directly computing $\nabla_{d_z} \mathcal{L}_c(z_T + d_z, \tau))$ is not trivial because the image generation process is iterative and involves typically hundreds of denoising steps, resulting in the vanishing gradient problem. In order to obtain a gradient for $d_z$, we introduce a residual connection from the input latent code $z_T$ into one of the intermediate time steps $t$ in the diffusion model, by adding the perturbation $d_z$ to the latent code $z_t^* = \omega z_t + (1 - \omega)d_z$. Note that we use a large $\omega = 0.9$ to not distort intermediate latent code (*i.e.*, when only adding $d_z$ to the intermediate layer, the output result is not changed, as demonstrated in Supp. B.6). Also note that the residual connection is removed after the optimization to generate the final result.

## 3.4 DESIGNING A ROBUST SURROGATE OBJECTIVE

A key component in our method is the design of an effective objective function that enables our optimization framework to discover failure modes of text-guided diffusion models. We propose to use discriminative image classifiers as a surrogate objective. But this comes at the risk of discovering false failures that attack the discriminative model, instead of attacking the generative model, as image classifiers have well-known failure modes themselves (Szegedy et al., 2013; Geirhos et al., 2019).

In our framework, an effective discriminator $\mathcal{D}$ needs to be able to tell whether the key object $[\mathtt{class}]$ is present in the generated image, while at the same time being highly robust. Therefore, we utilize an ensemble of commonly used discriminative models, such as Vision Transformer (ViT) (Dosovitskiy et al., 2020), Detectron2 (Wu et al., 2019), and the current SOTA method on ImageNet-C (Hendrycks & Dietterich, 2019) (*i.e.*, DAT (Mao et al., 2022)). We observe that even this ensemble of models fails to handle drastic style changes and is still sensitive to adversarial noise, leading to high false negative rates, as shown in our ablation (Sec. 4.2). Therefore, we further train a classifier that is biased towards shape cues by learning to classify ImageNet images that contain only the Canny edge maps (Canny, 1986). Finally, we build a robust discriminator $\mathcal{D}$ by summing the prediction probabilities of the key object class across all discriminators, including the edge-based classifier, which significantly enhances the success rate of SAGE (Sec. 4.2). We also train a binary classifier to distinguish distorted and natural images, and it is only used when searching for Type-2 failures in latent space. More details are provided in Supp. A.1.

## 4 EXPERIMENTS

In the following, we show the effectiveness of SAGE on three TDMs, *i.e.*, Stable Diffusion, DeepFloyd. and GLIDE. We also test on StyleGAN-XL to show its general applicability. Experiments involving LLaMA and experiments of baselines were conducted on a single A100 GPU, while other experiments were on a single V100 GPU. Hyperparameters and implementation details are listed in Supp. A.3.

**Baseline.** To the best of our knowledge, SAGE is the first method that is able to effectively search through the latent space and the human-understandable language space for failure cases. As we

Table 1: **Effectiveness of SAGE.** We report search success rate (SSR ↑), failure generation rate (FGR ↑), non-Gaussian rate (NGR ↓) and CLIP similarity (CLIPS ↑) under human (H) and automatic (A) evaluation. SAGE is able to efficiently search over all input spaces of text-guided generative models and find their weaknesses. All failure cases are verified by human annotators. In addition, our experiments on more advanced TDM, DeepFloyd, which uses LLM as text encoder, demonstrate that even if some issues are resolved in future models, SAGE will still be useful for finding new errors.

| Generator | Method | Latent space | | | Token embedding space | | | | Human-understandable text prompt space | | | |
|---|---|---|---|---|---|---|---|---|---|---|---|---|
| | | SSR(A) | SSR(H) | NGR | SSR(A) | SSR(H) | FGR(H) | CLIPS | SSR(A) | SSR(H) | FGR(A) | FGR(H) |
| SD V2.1 | Baseline | 64.3% | 12.5% | 75.0% | 0.0% | 0.0% | - | - | 9.6% | 6.4% | 35.1% | 29.5% |
| SD V2.1 | SAGE | 100.0% | 80.7% | 12.7% | 100.0% | 87.5% | 86.7% | 0.698 | 55.3% | 59.8% | 45.4 % | 67.5% |
| DeepFloyd | Baseline | 59.8% | 11.3% | 72.4% | 0.0% | 0.0% | - | - | 2.1% | 3.5% | 27.4% | 18.7% |
| DeepFloyd | SAGE | 100.0% | 74.6% | 14.1% | 100.0% | 84.2% | 82.5% | 0.610 | 47.5% | 52.1% | 49.3 % | 60.4% |
| GLIDE | SAGE | 100.0% | 88.9% | 8.6% | 100.0% | 88.0% | 94.7% | 0.677 | 60.9% | 70.7% | 34.2% | 76.4% |
| StyleGAN-XL | SAGE | 100.0% | 85.0% | 7.4% | - | - | - | - | - | - | - | - |

discussed before, previous methods only focus on language space, and they can only detect gibberish that is not interpretable. Therefore, we use standard PGD (for latent space), random sampling (for embedding space), and greedy search with LLaMA (for prompt space) as the baselines. Note that we applied several techniques to the baselines to enable running them on A100 GPUs (such as parameter freezing during gradient computation for PGD). See Supp. A.3 for details.

**Evaluation metrics.** *(i) To evaluate the failures of generative models*, we mainly consider two metrics: **search success rate (SSR)** and **failure generation rate (FGR)**. SSR refers to the percentage of instances where we find the failure cases within a fixed number of time steps. **FGR** is used for experiments involving random sampling of latent variables $z$: for each token/prompt we find, we randomly generate 100 images and report the percentage of failure images. Note that SSR is computed based on FGR. A search is considered successful if the FGR exceeds $80\%$ in the embedding space or $10\%$ when performed in the text space. *(ii) To evaluate the validity of latent variables $z$ and embeddings*, we consider two metrics: **Non-Gaussian Rate (NGR)** and **CLIP similarity (CLIPS)**. NGR refers to the number of latent codes $z$ that are unlikely to be sampled from a $\mathcal{N}(\mathbf{0}, \mathbf{I})$, *i.e.* they are outliers in a standard Gaussian distribution. To evaluate if a latent sample $z$ is an outlier, we use the Shapiro-Wilk test (Shapiro & Wilk, 1965), wherein a p-value below $0.05$ indicates that the hypothesis of following a standard Gaussian should be rejected. We also compute the mean and variance with a threshold of $0.05$ and $1.05$, respectively, to identify non-Gaussian samples. **CLIPS** measures the CLIP similarity between the embedding that includes the adversarial token with the one that excludes it. We only report the NGR, FGR, and CLIPS on successful cases.

**Evaluation of images.** One challenge we face is the lack of reliable metrics to evaluate whether an image is natural and relevant to the input prompt. The failure cases we find can be caused by the failure of generative models (true negative) or the failure of discriminators (false negative). Therefore, to better evaluate the generated images and also prove the ability of SAGE to find true failures of diffusion models, we perform both an **automated evaluation (A)** and a **human evaluation (H)**. In the **human evaluation**, three annotators evaluate if an image corresponds to an input prompt. They also evaluate whether the prompts discovered by SAGE are intelligible. In the **automated evaluation** process, we use the discriminator $\mathcal{D}$ in Sec. 3.4 to detect whether the key object is in the image.

## 4.1 Effectiveness of SAGE

We select 20 common object categories as the key objects for our experiments. We repeat each experiment 512 times and report the average performance in Tab. 1. We can see that our method is able to efficiently search through all input spaces of TDMs and automatically detect their weaknesses and failure cases. It's worth noting that **SAGE is able to find true failures of generative models**. For example on SD V2.1, $80.7\%$ of the failure examples we find are the failures of the generative models, which are validated by human annotators. We also show that **most failure cases SAGE finds are valid**. On SD V2.1, only $12.7\%$ of latent variables discovered by SAGE are probable outlier samples. In contrast, $75.0\%$ of failures found by the baseline (*i.e.*, PGD) are considered outliers, not to mention its notably low success rate of $12.5\%$ (resulting in a mere $3.1\%$ valid success rate). The embeddings we find also have high CLIP similarity compared with the original inputs. In addition, **even when searching over the discrete text prompt space, which no previous work has successfully done to find human-understandable prompts, our method achieves a high success rate, as verified by human annotators.** Most importantly, DeepFloyd utilizes a large language model (T5-XXL) as the text encoder, demonstrating significant enhancement in understanding human languages and alleviating several problems found in Stable Diffusion (see discussion in Sec. 5.1). However, our

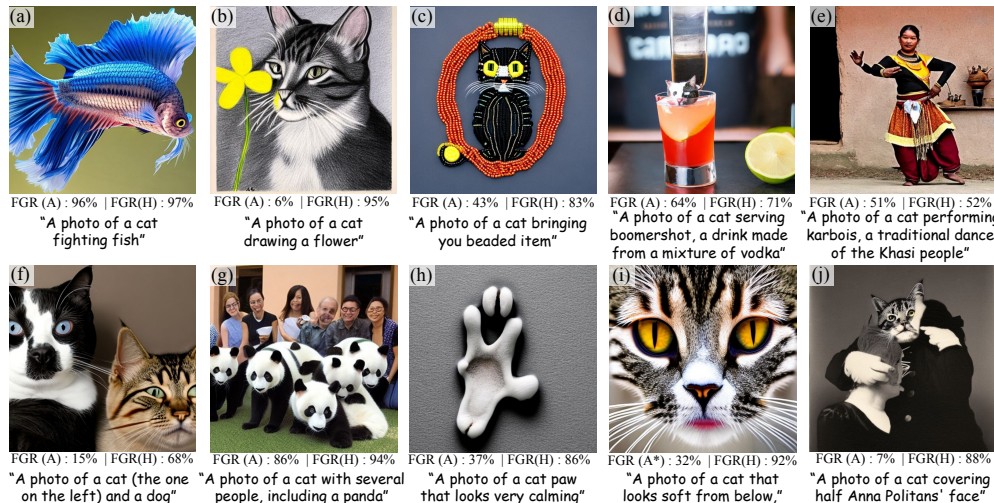

Figure 4: **Incomprehensible language prompts.** We show 10 representative examples here, and report the failure generation rate (FGR) under human (H) and automatic (A) evaluation. We only show results on 'cat' here and provide more results on different categories in Supp. C.

high search success rate on DeepFloyd (52.1%) shows that, **even if some issues are fixed in future models, SAGE will still be useful to find new errors and to help continually improve models.**

## 4.2 ABLATION STUDY ON DIFFERENT OPTIMIZATION TARGETS

Due to space limits, we only ablate optimization targets here. Ablations on the residual connection, denoising steps, text generation models, and hyper-parameters can be found in Supp. B.

We compare the failure images we find with and without an edge-based classifier (*i.e.* edge ViT) on SD V2.1 and repeat it 40 times. As shown in Fig. 3, the classifier on Canny edge maps is very important to remove the discriminator's bias toward textures and sensitivity to drastic style changes. We report the search success rate (SSR) on human evaluation (H) in the figure. It significantly improves the SSR(H) and enhances the true negative rate from 32.5% to 77.5%, demonstrating that it provides a more robust surrogate loss and reduces the gap between human and computer evaluation.

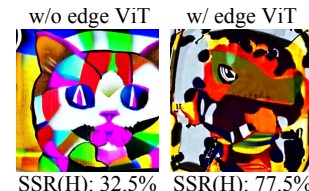

Figure 3: **Effect of edge ViT.**

## 5 FAILURE MODES OF TEXT-GUIDED DIFFUSION MODELS

### 5.1 NATURAL TEXT PROMPTS THAT ARE UNINTELLIGIBLE FOR DIFFUSION MODELS

Fig. 4 shows 10 representative examples of input texts that humans can easily understand but SD V2.1 cannot, revealing several existing limitations of TDMs: Current models are good at understanding objects, but may be confused by specific actions ('fighting' in Fig. 4 a and 'tracking' in Fig 1 a), and may struggle to understand the relationship between nouns and verbs (Fig. 4 b), nouns and adjectives (c), subjects and objects (d), and words with salient features (e). In addition, they often do not comprehend compositions (f), numbers (g), parts (h), viewpoints (i), and fractions (j).

It is noteworthy that in previous work, people tried to *manually* generate prompts to test TDMs, and they found some failure cases that align with our findings. But they mainly focus on the composition of multiple objects (Gokhale et al., 2022; Conwell & Ullman, 2023), which is the most evident error, accounting for just a small part of our finding (*i.e.*, Fig. 4 f). **We also find many interesting errors that have never been reported before.** For example, we observed that certain actions may easily fool the TMDs. Specifically, "A photo of a cat and fish" can typically give good results, but if we substitute "and" with "fighting", 97% of the generated images do not contain a cat. The same problem happens when replacing "and" with "tracking" in "cat and bear" (no cat in 96% of the images), and replacing "and" with "chasing" in "cat and dog" (no cat in 66% of the images). Other previously unexplored failure models include the object-merging problem (Fig. 4 d), the existence of salient words (Fig. 4 e), and so on. Please see Supp. B.2 for a detailed examination of each failure mode.

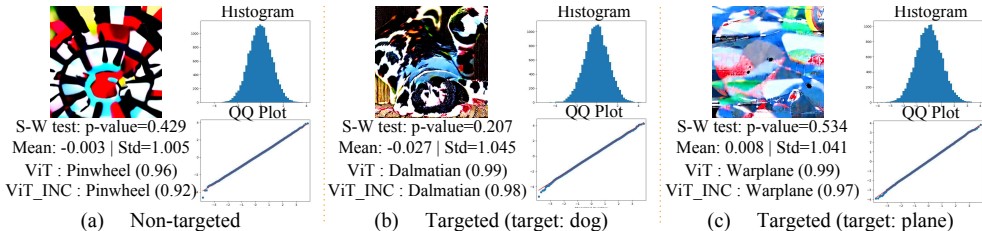

Figure 5: **Distorted images with non-outlier latent variables.** We show three examples generated by the latent variables we find (prompt: "A photo of a cat"). For each example, we provide the generated image, the distribution information of the corresponding latent variable, and the corresponding classifier score of SOTA models on ImageNet and ImageNet-C.

**Exploring deeper causes and possible solutions.** For each failure mode, we study its deeper causes by using LLaMA to modify the adversarial prompts in three different manners, and we also compared the failure cases and failure rates of SD V2.1 and DeepFloyd of each failure mode (results and details are reported in Tab. 3 and Fig. 8 in Supp. B.2). The former employs CLIP as a text encoder, whereas the latter utilizes T5-XXL. By comparing their failure cases, we find two primary sources of failures: the comprehension ability of text encoders, and the expression ability of diffusion models. Specifically, we find that DeepFloyd shows more robustness to noun-verb (b), noun-adjective (c), subject-object (d) relationships, words with salient features (e), and compositions (f). Through a detailed examination of individual failure cases, we think the failure of misrepresenting the word relations and object relations can be alleviated by a better language model with more fine-grained understanding. In addition, DeepFloyd shows a better understanding of numbers (g) and fractions (j), but it still cannot always describe these concepts well. For example, we find that even if the text encoder understands fractions, the diffusion model may still fail to generate the correct objects. Also, we find that the existing diffusion models struggle to express object viewpoints (i) and parts (h). Finally, we also find that all models show a high failure rate on certain actions, but interestingly, the failure reasons seem to be different. Specifically, CLIP-based methods tend to generate nice images, yet they often lack the key object, implying potential disruptions in the text encoder caused by these actions. While a portion of images (33.2%) generated by DeepFloyd include all key objects, these actions cause a strange or distorted appearance of the objects. This implies that, despite understanding some actions, the diffusion models still struggle to convey the scene under these actions.

Due to space constraints, we only detail one previously unexplored finding as an example, and summarize the underlying causes and potential solutions very briefly. For a more comprehensive explanation and discussion of each failure mode, please refer to Supp. B.2.

## 5.2 TYPE-1: LATENT SAMPLES THAT LEAD TO DISTORTED IMAGES

One category of frequent failures in the latent space is distorted images as shown in Fig 5. The latent variables used to generate these images are **non-outlier samples** of $\mathcal{N}(\mathbf{0}, \mathbf{I})$, which is demonstrated by the histogram, QQ plot, mean, standard deviation, and Shapiro-Wilk test. We also find that these distorted images can be generated with random sampling. Furthermore, we demonstrate that these failure examples are not isolated spots in the latent space; rather, they exist as connected regions, wherein any sample will generate similar distorted images (see Supp. B.3). It shows that these distorted images can be generated with non-zero probability under Gaussian sampling according to the probability density function. These findings suggest that the latent space is not well-structured in some parts and hence shows potential to be improved. It also illustrates the representation bias present in both generative and discriminative models, which diverges from that of human vision (see Supp. B.4).

## 5.3 TYPE-2: LATENT SAMPLES THAT DO NOT DEPICT THE KEY OBJECT

Fig. 6 illustrates another category of frequent failures in the latent space which generates images that do not depict the key objects but associated backgrounds. Also here, the latent variables used to generate these images are **non-outlier samples** of $\mathcal{N}(\mathbf{0}, \mathbf{I})$. We have also observed that for some particularly brittle categories found by our method,

Table 2: **Model stability.**

|  | SD V2.1 | SD V1.5 | GLIDE |
| --- | --- | --- | --- |
| Region Size | 0.024 | 0.037 | 0.060 |
| Failure Rate | 2.7% | 9.2% | 50.5% |

such as cat/car for SD V1.5 and umbrella/basketball for V2.1, even random sampling can produce such unrelated images. Tab. 2 compares the normalized size of these failure regions and the probability of

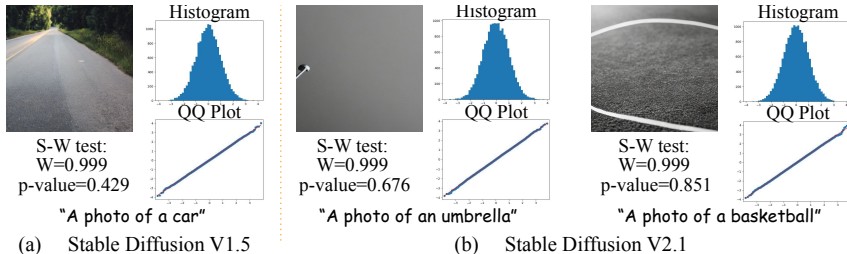

Figure 6: **Associated backgrounds with non-outlier latent variables.** We find non-outlier latent variables that do not depict the key objects but associated backgrounds. We provide the generated image and the distribution information of the corresponding latent variable.

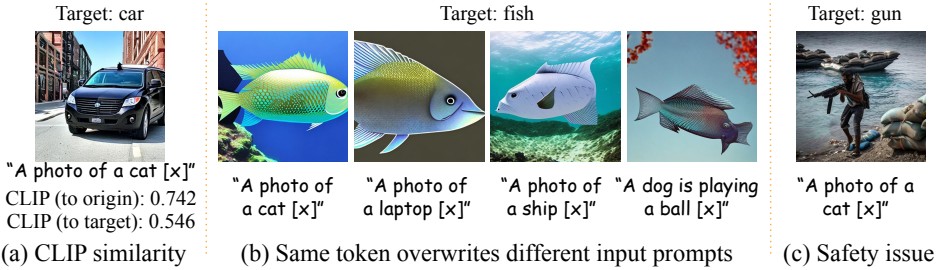

Figure 7: **Token embeddings that overwrite the input prompts.** We find token embeddings that can overwrite any input prompts and generate images with only the target objects.

unrelated images being generated under random sampling, which we define as model stability. For five of the brittle categories SAGE finds, even with the simplest template, 9.2% and 50.5% of randomly generated images are actually irrelevant when using SD V1.5 and GLIDE, respectively. It shows that the size of Type-2 failure indicates the model stability, and SAGE provides a highly efficient way to evaluate what object categories are not sufficiently well represented (more in Supp. B.5).

## 5.4 TOKEN EMBEDDINGS THAT OVERWRITE THE INPUT PROMPTS

We have discovered that by appending a single token embedding, denoted by [x], to the original input prompt, we can overwrite the prompt and generate an image of a target object, without changing the CLIP similarity score significantly. Note that a similar phenomenon was also reported before (Zhuang et al., 2023; Mao et al., 2022), but we are the first to find adversarial tokens have little effect on CLIP score, implying that we are indeed attacking the diffusion process. However, the tokens they found do not show such properties. For example in Fig. 7 (a), the CLIP similarity score between the new input ("A photo of a cat [x]") and the original input ("A photo of a cat") is still higher than the score between the new input and the target ("A photo of a car"), (0.742 *v.s.* 0.546). However, the new input will always generate a car. Note that directly appending a "car" to "A photo of a cat" will generate images with both car and cat with a large probability.

More importantly, compared with prior findings, we have revealed the existence of universal token embeddings. As illustrated in Fig. 7 (b), certain token overwrites input text prompts for different key objects, and even prompts with different language patterns, causing the model to consistently generate images with only the target object. We have also found token embeddings that can generate safety-critical materials, such as weapons, drug use, nudity, blood, *etc*. (Fig. 7 c). This type of failure underscores the sensitivity of current language representations. Specifically, it exhibits a notable bias towards the final word of the input, predominantly focusing on the noun/object, and is easily misled by particular words, forming the foundation of these universal token embeddings. We firmly believe these issues should be solved and can be solved by, for example, a better training strategy of CLIP with more fine-grained information, to get more robust and reliable TDMs.

## 6 CONCLUSION

In this work, we present SAGE, the first method to automatically study the reliability of TDMs. It implements a gradient-guided search over the discrete prompt space and the high-dimensional latent space to discover failure cases in image generation. By studying TDMs with SAGE, we reveal four typical failure modes that have not been systematically studied before.

## ETHICS STATEMENT

Diffusion Models have many positive qualities but can be misused, *e.g.*, to create Deep Fakes. Our work is an attempt to understand their weaknesses, *e.g.*, they can be attacked to generate a photo of safety-critical materials even when the prompt is "A photo of a cat". Understanding the weaknesses is a necessary, but not sufficient, pre-requisite to prevent misuse.

## ACKNOWLEDGEMENT

We would like to thank Yujia Wang, Yanqing Zhang, William Tang, Waytt Song, and Zehao Zhong for their help in evaluating the images. We would also like to thank Artur Jesslen and Yujia Wang for their proofreading and helpful comments. This work was done in part during an internship at Bytedance. Alan Yuille acknowledges support from the ONR N00014-23-1-264 and Army Research Laboratory award W911NF2320008. Adam Kortylewski acknowledges support via his Emmy Noether Research Group funded by the German Science Foundation (DFG) under Grant No. 468670075.

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

APPENDICES

Here we provide details and extended experimental results omitted from the main paper for brevity. Sec. A gives more details of SAGE, including loss functions and implementation details. Sec. B contains the extended experiments and in-depth discussions regarding the reported failure modes. Sec. C provides more representative text prompts that diffusion models cannot understand, as well as five corresponding generated images.

## A MODEL DETAILS

### A.1 LOSSES

SAGE uses an assembly of discriminative models (multiple classifiers and one object detector) to build a robust discriminator. Assume $q$ denotes the number of the discriminative models, and $\mathcal{P}_i(I)$ denotes the output probability score of the key object (*i.e.*, positive) given by the discriminative model $\mathcal{C}_i$ when image $I$ is provided, then:

$$\mathcal{D}(I) = \sum_{i=1}^{q}(1 - 2\mathcal{P}_i(I)) \tag{5}$$

**Targeted optimization.** We can also optimize the surrogate such that the output of the diffusion model will depict a specific target object that is different from the key object in the text prompt. To achieve this, we modify the optimization target by not only minimizing the score of the key object, but also maximizing the score of the target object with a cross-entropy loss. We further use a pretrained CLIP model (Radford et al., 2021) to maximize the cosine similarity between the generated images and the prompt of the targeted object (*i.e.*, "A photo of a `[target]`").

Specificaly, when a target object $y$ is provided, the output of the discriminator $\mathcal{D}$ is:

$$\mathcal{D}(I) = \sum_{i=1}^{q}(1 - 2\mathcal{P}_i(I)) - \lambda_1 \sum_{i=1}^{q}(\text{CEL}(\mathcal{C}_i(I), y)) + \lambda_2 \mathcal{S}(I, \text{T}(y)) \tag{6}$$

where $\lambda_1 = 1$, $\lambda_2 = 2$, CEL$(\cdot)$ is the cross-entropy loss, $\mathcal{S}(\cdot)$ computes the cosine similarity between two CLIP embeddings, and T$(y)$ is the input prompt of the target object $y$.

### A.2 PSEUDOCODE

We provide the pseudocode of the gradient-guided search for text prompts in Algo. 1.

---

**Algorithm 1** : Gradient-guided search for text prompts

---

**Require:** $L$: language model for text generation (LLaMA), $\mathcal{G}$: text-to-image generator, $\mathcal{D}$: robust discriminator, $\tau$: CLIP text encoder, $m$: maximum text length, $n$: number of inner iteration.
    Initialize template prompt $P = $ "A photo of a `[Class]`"
    **for** $t = 1, 2, ..., m$ **do**
        Using $L$ to generated $k$ possible words `[c]`, and compute token embeddings $\tau_{[c]} = \tau(L(P))$
        Compute the token embedding of $P$ : $\tau_p = \tau(P)$
        Initialize latent code $z \sim \mathcal{N}(0, I)$
        Random initialize words $\tau_{[x,y]}$ based on the range of $\tau_{[c]}$.
        **for** $d = 1, 2, ..., n$ **do**
            Update current token $\tau' = \text{concatenate}(\tau_p, \tau_{[x,y]})$
            Compute loss $\mathcal{L}_d(z, \tau') = -\mathcal{D}(\mathcal{G}(z, \tau')) + \lambda\mathcal{S}(\tau_{[x]}, \tau_{[c_n]})$
            Update $\tau_{[x,y]} \leftarrow \tau_{[x,y]} + r \cdot \alpha \cdot \text{sgn}(\nabla_{\tau_{[x,y]}} \mathcal{L}_d(z, \tau'))$
        **end for**
        Find the closest candidate `[w]` in `[c]`: `[w]` $= \arg\min_n \mathcal{S}(\tau_{[x]}, \tau_{[c_n]})$
        Update prompt $P \leftarrow \text{append}(P, $ `[w]` $)$
    **end for**

---

### A.3 Implementation Details

**SAGE.** The vision transformers we use to build the discriminator are based on the Timm Library (Wightman, 2019). We use the LLaMA 7B (Touvron et al., 2023) model to generate texts. For the search over the *latent space*, we search for at most 500 iterations with $\alpha = 5 \times 10^{-2}$ and constrain the perturbation $d_z$ to be in the range $d_z \in [-1, 1]$. We add residual connection to denoising step $t = 20$ with weight $\omega = 0.9$. For the search over the *token embedding space*, we search for at most 250 iterations with $\alpha = 1 \times 10^{-3}$ and constrain the embedding $\tau$ to be in the range $\tau \in [-2.5, 2.5]$. We use the gradient from denoising step $t = 5$. For the search over the *prompt space*, we search for at most 100 iterations for each word, with $\alpha = 5 \times 10^{-2}$, $\lambda = 0.1$ and constrain the embedding $\tau$ to be in the range $\tau \in [-2.5, 2.5]$. We use the gradient from denoising step $t = 5$. For each input prompt, LLaMA gives at least $k = 100$ candidates and the maximal length of the generated text is $m = 10$.

**Baseline.** For the search over the *latent space*, we employ a standard adversarial attack, *i.e.*, projected gradient descent (Madry et al., 2017), as our baseline method. PGD is chosen because our method is based on it and it is widely used in the field. Given that attacking the latent space requires backpropagating gradients from the output images to the very deep latent space, we use FP16 precision and employ parameter freezing during gradient computations to fit the algorithm into an A100 GPU with 80G of RAM.

For the experiment regarding the *token embedding space*, we adopt random sampling as the baseline, since the most closely related method (Wen et al., 2023) involves minimizing the CLIP similarity between text and image feature vectors, while our method aims is designed to manipulate the token embedding space of TDMs without significantly changing the CLIP similarity.

For the search over the *text prompt space*, we adopt greedy search as the baseline. Similarly to SAGE, given an input text "A photo of a `[class]`", we utilize LLaMA to generate 100 candidates that can be added to the input prompt. Then, instead of using the proposed searching policy, we traverse all candidates to find the word with the lowest ViT score. It is then used as the next word (*i.e.*, `[w]` in the main paper) that is added to the current input prompt. Similarly, we repeat the entire process for a maximum of $m = 10$ times or until the sentence is completed.

## B Extended Experiments and Discussions

### B.1 Experimental Evaluation Details

**Object categories.** In Sec. 4.1 of the main paper, to demonstrate the effectiveness of SAGE, we select 20 common object categories as the key objects. They include "cat", "dog", "bird", "fish", "horse", "car", "plane", "train", "ship", "laptop", "chair", "bike", "television", "bear", "monkey", "sheep", "cow", "cock", "snake", and "butterfly". For each category, we consider all relevant subcategories in ImageNet-1k as the correct category when building the optimization targets. For example, when "cat" is the key object, all categories that are relevant to "cat" in the ImageNet-1k dataset, such as "tabby" (n02123045), "tiger cat" (n02123159), *etc.*, are considered as the correct category labels.

**Human evaluation.** For human evaluation, we have 7 evaluators (3 females and 4 males). They are PhD or master's students in three different institutions, aging from 22 to 28, studying computer science (3), mechanical engineering (2), mathematics (1), and international economics (1). We randomly designate three evaluators for each generated image. Their task is to assess whether the images accurately depict the input prompts, and assign scores on a scale of 1 to 5. A score of 1 indicates very poor alignment with no related contents while 5 means a precise depiction. The final score for each image is the average of their ratings. Then we consider the score greater and equal to 3 as indicative of a correct image. They also assess whether the prompts discovered by SAGE are natural and intelligible. In addition, we provide the statistics for the scores obtained for each image in Tab. R-1. Notably, the ratings exhibit good consistency across the majority of the images.

**Automatic evaluation.** For automatic evaluation, we use the discriminator $\mathcal{D}$ to detect whether the key object is in the image. However, language models can generate prompts involving viewpoints, complex actions, and other concepts that discriminators cannot detect. To alleviate this challenge, we employ two approaches. *(i)* We incorporate symmetry detection in the evaluation step. We have observed that if the model fails to understand the prompt, it may produce highly symmetrical images of the key object's face. Note that we only use it as a supplement and it is denoted as (A*) when in

Table R-1: **Score statistics for human evaluation**

| Variance | $\leq 0.222$ | 0.222-1 | $\geq 1$ |
|---|---|---|---|
| Proportion | 76.2% | 23.4% | 0.4% |

Table 3: **Failure generation rate per category.** We compare the average FGR(H) of each failure category on Stable Diffusion V2.1 and DeepFloyd.

| | (a) Motion | (b) Noun-verb | (c) Noun-adjective | (d) Subject-object | (e) Salient feature |
|---|---|---|---|---|---|
| SD V2.1 | 66.3% | 90.4% | 91.5% | 78.9% | 73.3% |
| DeepFloyd | 50.1% | 26.5% | 31.4% | 42.6% | 38.6% |

| | (f) Composition | (g) Numbers | (h) Parts | (i) Viewpoints | (j) Fractions |
|---|---|---|---|---|---|
| SD V2.1 | 80.7% | 95.5% | 87.3% | 89.5% | 71.8% |
| DeepFloyd | 30.5% | 76.3% | 82.4% | 78.8% | 63.4% |

use. *(ii)* For prompt evaluation, we loosen the threshold of successful search from 80% FGR to 10%. It is based on our finding that when searching for prompts, if the model gives several images without the key object, many of the other images actually fail to capture the key concept of the prompt.

## B.2 MORE DISCUSSIONS ON INCOMPREHENSIBLE PROMPTS FOR DIFFUSION MODELS

In this section, we discuss in detail the different failure modes uncovered by SAGE. Specifically, we delve into the underlying factors behind each failure mode through a comprehensive approach, including (1) employing LLaMA to generate synonyms of the keywords in adversarial prompts (*i.e.*, the keywords identified by SAGE), (2) switching the subject and object of the adversarial prompts when the keyword is a predicate, (3) making alterations to the adversarial prompts while retaining the core concepts, and (4) conducting comparative analysis of failure cases and failure rates for each failure mode between CLIP-based TDM (SD V2.1) and LLM-based TDM (DeepFloyd). After that, we discuss possible solutions for each failure mode based on our experiment results. The comparisons between SD V2.1 and DeepFloyd are provided in Tab. 3 and Fig. 8.

The discussion on each failure mode is presented as follows:

(a) **Specific actions:** One interesting failure that has not been previously recognized is that specific actions may easily fool the TMDs. For example, on SD V2.1, "A photo of a cat and fish" can typically give good results, but if we replace "and" with "fighting" into "A photo of a cat fighting fish", 97% of the generated images do not even contain a cat. Likewise, we observe that "A photo of a cat chasing dogs" causes a 66% failure generation rate. If we substitute the keyword "chasing" with "playing with", then the generated prompt "A photo of a cat playing with dogs" exhibits a significantly reduced failure generation rate of 29%. We observe three possible causes of this failure. The first reason is the incorrect pairing of specific verbs with nouns. For example, in SD V2.1, "cat fighting fish" consistently gets interpreted as a type of fish (*i.e.* fighting fish). This wrong interpretation does not occur in SD V1.5, GLIDE, or DeepFloyd. The second reason is that some scenarios described by the action of the subject and object are less common in real life (*e.g.* "cat teaches horse how to walk" in Fig. 8, "a bird running back" in Fig. 16), making it challenging for the text encoder to comprehend and the diffusion model to generate (*i.e.*, out-of-distribution cases). The third reason is that representing certain actions requires the deformation of target objects, which is also a challenge for the diffusion model. For instance, we find that generating the "chasing" action on a cat is challenging, often resulting in images depicting an unidentifiable object chasing a dog.

(b) **Noun-verb relationships:** Another common failure observed in CLIP-based TDMs related to actions is the incorrect interpretation of noun-verb relationships. For instance, the phrase "a cat drawing a flower" is consistently misinterpreted as "a drawing of a cat and a flower". In contrast, this problem is significantly mitigated in the LLM-based model (DeepFloyd). We think this issue may be mainly due to the training strategy of CLIP. Since it is trained on extensive image-text pairs with contrastive loss, it excels at capturing objects, and it can also capture certain verbs

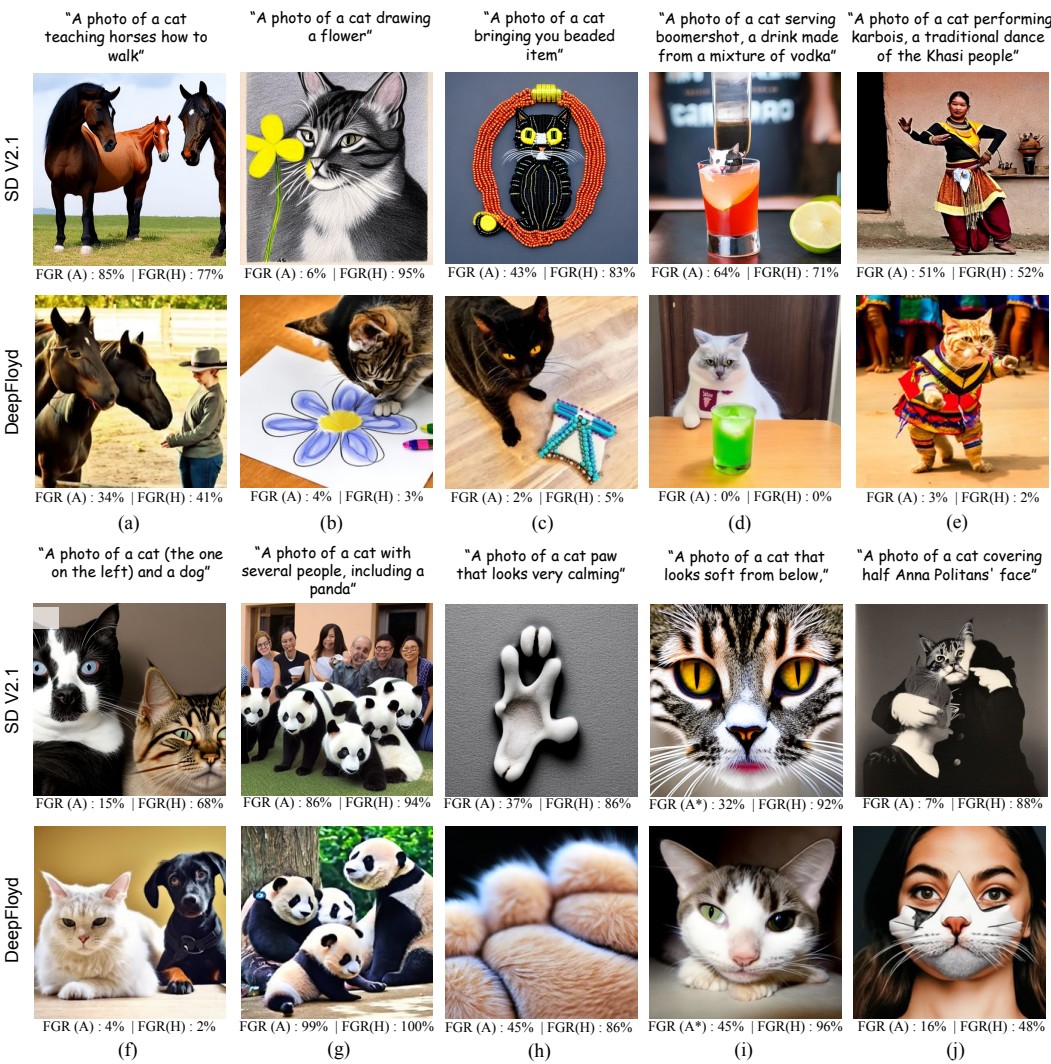

Figure 8: **Comparing Stable Diffusion V2.1 and DeepFloyd performance on incomprehensible prompts.**

occasionally, but it shows notable shortcomings in understanding their relationships. We believe such weakness can be alleviated by a better training strategy that, for example, incorporates more fine-grained information.

(c) **Noun-adjective relationships:** This failure is similar to the noun-verb relationships. Specifically, CLIP-based TDMs struggle to understand the relation between nouns and adjectives, especially when there are multiple objects. A typical failure is that an adjective is acted on wrong objects. This problem is also mitigated in the LLM-based model (DeepFloyd). We believe such weakness can also be alleviated by a better training strategy with more fine-grained information.

(d) **Subject-object relationships:** Another interesting finding is that, sometimes, CLIP-based methods attempt to merge two or more different objects into one strange object that has features from all objects involved. For example, SD V2.1 may generate a bottle of wine with a cat's head (Fig. 8 d), cat-like cookies(Fig. 13), or a corgi with a bird-like beak (Fig. 16). This phenomenon is less prevalent in LLM-based TDMs. We attribute this difference primarily to the training process of CLIP, where a lot of images describe only one object per image. Therefore, CLIP may have the tendency to merge all feature vectors onto one single object, especially when a complex prompt is provided. It's important to note that while LLM exhibits greater robustness,

it still shows a high failure generation rate on this problem. It shows that LLM is also sensitive to subject-object relationships, and may not fully grasp the true meaning of the input prompts.

(e) **Words with salient features:** In addition to the merging of different feature vectors onto one object, CLIP-based methods may also be influenced significantly by the word with salient features, such as "Khasi people" in Fig. 8 (f) and "Korean" in Fig. 15. And this problem is more serious when the words with salient features are at the end of the input. Compared with the LLM-based method, CLIP is more likely to exhibit bias towards the last word of the inputs, and words with salient features. Note that LLM is also sensitive to this problem. Since CLIP may combine all features into one object, the TDMs based on it can be easily disturbed by certain words with salient features. We believe these can be alleviated by a better training strategy for CLIP.

(f) **Composition of multiple objects:** This is the main failure of TDMs that has been discussed before (Gokhale et al., 2022; Marcus et al., 2022; Conwell & Ullman, 2023). To address this issue, people either explicitly compose different diffusion models (Liu et al., 2022; Du et al., 2023; Kumari et al., 2023; Bar-Tal et al., 2023) or use guidance to compose multiple objects into one image (Epstein et al., 2023; Chefer et al., 2023; Feng et al., 2022). In our experiments, we observe that the LLM-based method demonstrates a superior capacity for understanding and generating complex compositions. This suggests that the deeper cause of this problem lies within the text encoder of TDMs. CLIP often lacks the capability to fully understand the composition of multiple objects.

(g) **Numbers:** Our experiments show that all TDMs display poor proficiency in understanding numerical concepts. Although DeepFloyd demonstrates better a understanding of numbers, it still encounters challenges in consistently interpreting these concepts accurately. And this issue becomes more serious when numerical concepts are combined with other challenges, such as compositions. For example, an input prompt like "several people and a panda" may lead to images with only pandas. (*e.g.*, Fig. 8 f)

(h) **Parts:** Although TDMs excel in generating objects, they show notable weaknesses when it comes to generating object parts. For example, they struggle to generate "cat paw", and an input prompt like "cat feet" will give images of fluffy human feet instead. However, if we take a closer look at the generated images, it becomes evident that diffusion models do understand what they are asked to generate. Nonetheless, it cannot describe the object parts very well. Furthermore, it also struggles to understand the relationships between parts and the entire object, sometimes generating images of an isolated, floating fluffy part of an unidentified animal.

(i) **Viewpoints:** Another failure in generating individual objects lies in the struggle to understand different object viewpoints. When generating objects, particularly animals, the majority of images depict animals/objects facing forward. Moreover, you cannot alter the viewpoints by providing explicit descriptions such as "looking from below". Interestingly, in some cases, such a description exacerbates the problem, resulting in more images showing a front view of the target object with only a highly symmetric face (*e.g.*, Fig. 8 i).

(j) **Fractions:** Fractions also pose a large challenge for all TDMs, and we have identified two primary types of failures. For CLIP-based models, they may struggle to understand the concept of fractions. For example in Fig 8 (j), they may not understand what is "half of an object". For DeepFloyd, the utilization of LLM enables a better understanding of the concept of fractions, often resulting in images with a half-covered face. However, the diffusion process itself appears to lack knowledge of the entire object. Therefore, it may often generate images with a cat face sticker affixed, or just a floating cat face, rather than a cat with a realistic body.

### B.3 Distorted Regions of Type-1 Failure

We can search the space around the latent variables of type-1 failure for distorted regions. We adopt the failure region searching method in PoseExaminer (Liu et al., 2023). Specifically, it provides an intuitive way to compute the final boundary in a high-dimension system under certain constraints, in which they gradually expand the upper and lower boundary of each dimension. We follow their method, and starting from the latent variable we find, we gradually expand the boundary until it becomes an outlier or no longer causes the model to fail. Our experiments show that for each latent variable, we can find connected regions surrounding it, wherein any sample will generate similar distorted images (Fig. 9 a). It shows that these latent variables can be generated with meaningful

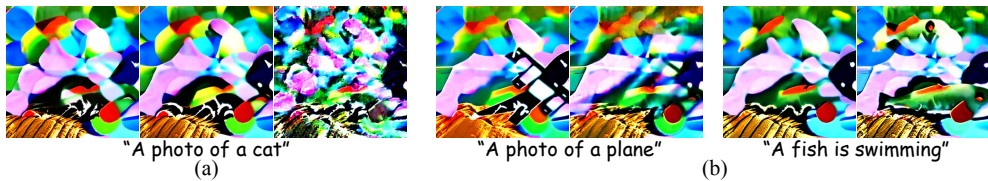

Figure 9: **Distorted Regions of Type-1 Failure.**

Table 4: **Type-2 Failure and Model Stability.**

|  | SDM V2.1 | SDM V1.5 | GLIDE |
|---|---|---|---|
| Search Success Rate (A) | 32.0% | 41.5% | 96.5% |
| Failure Region Size | 0.024 | 0.037 | 0.060 |
| Failure Generation Rate (random sampling) | 2.7% | 9.2% | 50.5% |

probability under Gaussian sampling. Since the probability of a range of values in a Gaussian distribution is not zero and can be computed by the probability density function. Interestingly, we also observe that these regions produce distorted images even when subjected to different text inputs (Fig. 9 b).

### B.4 INTERPRETING TYPE-1 FAILURE WITH CLASSIFIERS

Interestingly, when provided with a target class during optimization, we are capable of generating distorted images that can be interpreted as the target category with high confidence scores by both a standard ViT and a robust ViT on ImageNet-C, despite being unintelligible to humans. This indicates that both types of models have representation biases that differ from those of human vision, and Type-1 errors might also be useful to provide further insights into the decision process and representation biases of classifiers. For example in Fig. 5(b), we observe some white texture with black spots, which is actually one distinctive key feature of Dalmatians.

### B.5 TYPE-2 FAILURE AND MODEL STABILITY

The stability of generative models is a crucial factor in evaluating their performance. In our study, we find that SAGE can be used to stress test their stability by searching for type-2 failures for specific categories.

Tab. 4 evaluates the relation between the search success rate (SSR) of type-2 error, the normalized failure region size, and the quality of diffusion models under random sampling. We randomly generate $5000 \times 5$ images for *5 common but fragile object categories* and compute the failure generation rate of random sampling (FGRrs), and then compare it with the SSR. We observe a clear relation between the stability of diffusion models and the SSR reported by our method, demonstrating the potential of our method for evaluating the model stability. Note that the 5 object categories are different from those used in Sec. 4.1. We use "umbrella", "basketball", "car", "unicycle", and "alp" here. These five categories are also automatically found by SAGE. Specifically, we run SAGE on all 1K ImageNet-1k categories and then rank the categories according to the iteration step for convergence to get a list of fragile objects. Then we select five common but fragile objects.

### B.6 RESIDUAL CONNECTION DOES NOT DISTORT THE INTERMEDIATE LATENT VARIABLE

Residual connection (RC) is used to back-propagate the gradient from the output to the perturbation ($d_z$) of latent variable ($z$). Although it is only added during optimization and is removed after the optimization to generate the final result, we still want to make sure this connection does not distort the intermediate latent variable ($z_t$). Fig. 10 shows the images generated with and without RC, as well as the corresponding probability score of the key object (*i.e.*, $\mathcal{P}(I)$). RC only has a negligible influence on both the generated images and the resulting classification outcomes.

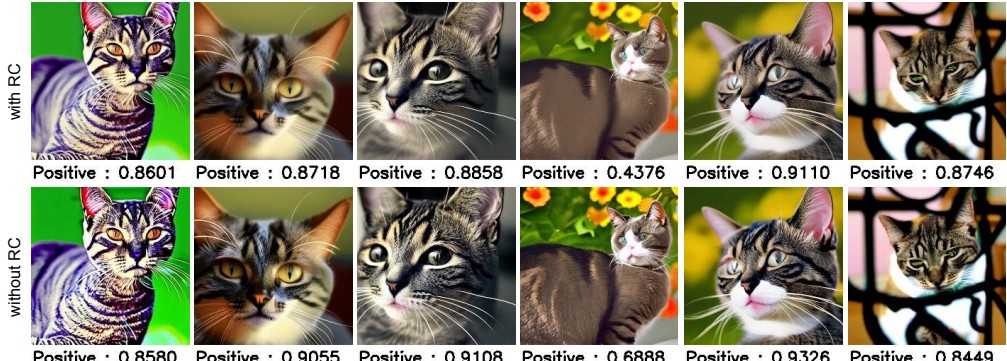

Figure 10: **Residual connection does not distort the intermediate latent variable.**

Table 6: **Ablations on different denoising steps.**

|        | 1     | 5     | 10    | 20    | 30    | 40   |
|--------|-------|-------|-------|-------|-------|------|
| SSR(A) | 100%  | 100%  | 100%  | 100%  | 100%  | 86%  |
| SSR(H) | 71%   | 69%   | 74%   | 76%   | 75%   | 78%  |

We compare $512 \times 500$ images and report the difference of the classification results between images with ($I^w$) and without ($I^{w/o}$) residual connection in Tab. 5. We compare two metrics: **Difference Rate (DR)**, which compares the percentage of the image pairs ($I^w$ and $I^{w/o}$) that have different Top-1 category prediction, and **Relative Change of Probability Score (RCPS)**, which compares the relative change of the probability score between the image pairs. Specifically, assume we have $n = 512 \times 500$ images, then:

Table 5: **Influence of RC.**

| DR    | RCPS |
|-------|------|
| 0.02% | 4.7% |

$$\text{DR} = \frac{\#(\text{Top-1}(I^w) \neq \text{Top-1}(I^{w/o}))}{512 \times 500} \times 100\% \tag{7}$$

$$\text{RCPS} = \sum_{i=1}^{n} \frac{|\mathcal{P}(I_i^{w/o}) - \mathcal{P}(I_i^w)|}{\mathcal{P}(I_i^w)} \times 100\% \tag{8}$$

### B.7 ABLATION OF RESIDUAL CONNECTION ON DIFFERENT DENOISING STEPS

In this section, we study the effect of adding residual connections to different denoising steps. To avoid running out of CUDA memory, all experiments in this section are performed on an A100 GPU with 80GB RAM. We consider the search over the latent space and add the residual connection to $t = 1, 5, 10, 20, 30, 40$ step. We repeat each experiment 100 times and report the results in Tab. 6. We can see that the gradient from the last few diffusion steps is more likely to change the style of the image while keeping the outline of the object, which slightly reduces the search success rate under human evaluation, *i.e.* SSR(H). However, the gradient from the deeper steps suffers from the gradient vanishing problem, which reduces the search success rate under automatic evaluation, *i.e.* SSR(A). We use $t = 20$ for searching over the latent space, and $t = 5$ for the prompt embedding space.

### B.8 ABLATION ON DIFFERENT TEXT GENERATORS

In addition to LLaMA (Touvron et al., 2023), we also consider GPT-2 (Radford et al., 2019) and T5 (Raffel et al., 2020) as the text generator and compare the results in Tab. 7. For each experiment,

Table 7: **Ablations on different text generators.** H* stands for human evaluation without evaluating the validity of generated prompt.

|         | SSR(A) | SSR(H*) | SSR(H) | FGR(A) | FGR(H) |
|---------|--------|---------|--------|--------|--------|
| LLaMA 7B | 50.8%  | 58.6%   | 56.3%  | 42.8%  | 60.1%  |
| GPT-2   | 49.2%  | 60.5%   | 48.0%  | 44.1%  | 64.5%  |
| T5      | 53.1%  | 57.4%   | 49.2%  | 38.9%  | 59.4%  |

Table 8: **Generating general prompts from scratch.** When being allowed to generate more general prompts without template, the attack success rate is even higher.

| | SSR(A) ↑ | SSR(H) ↑ | FGR(A) ↑ | FGR(H) ↑ |
|---|---|---|---|---|
| Template is added | 55.3% | 59.8% | 45.4 % | 67.5% |
| Template is removed | 69.7% | 75.2% | 60.2 % | 77.1% |

| (a) | (b) | (c) | (d) | (e) |
|---|---|---|---|---|
| FGR (A) : 95% \| FGR(H) : 99% | FGR (A) : 38% \| FGR(H) : 97% | FGR (A) : 73% \| FGR(H) : 95% | FGR (A) : 51% \| FGR(H) : 74% | FGR (A) : 56% \| FGR(H) : 65% |
| FGR (A) : 96% \| FGR(H) : 97% | FGR (A) : 6% \| FGR(H) : 95% | FGR (A) : 43% \| FGR(H) : 83% | FGR (A) : 64% \| FGR(H) : 71% | FGR (A) : 51% \| FGR(H) : 52% |
| "A cat fighting fish" | "A cat drawing a flower" | "A cat bringing you beaded item" | "A cat serving boomershot, a drink made from a mixture of vodka" | "A cat performing karbois, a traditional dance of the Khasi people" |

Figure 11: **Removing template "A photo of a** `[class]`**".** We visualize the generated images when the template is removed. We observe a rise in the Failure Generation Rate when the template is removed (white row), in comparison to prompts with the template (grey row).

we repeat 256 times and report the average performance. We can see that if we do not consider the validity of the generated prompt, all three methods achieve comparable performance. However, when considering the validity, LLaMA gives a slightly better performance. Note that their validity also depends on the number of candidates $k$. Larger $k$ values will also result in more invalid prompts. In general, the text generator only provides possible candidates for the next word. Therefore, any text generation model can be used here.

## B.9 GENERAL USE OF TEXT PROMPTS

In our experiments, we use the standard template "A photo of a `[class]`" because many TDMs are trained on it and it is often the best template in terms of image generation (Radford et al., 2021). Our goal is to highlight that failure cases occur even for prompts that are supposed to work the best. When simply removing the pre-defined template from our found prompts, we observe similar or even higher failure generation rates for all prompts (Fig. 11), proving their generalization ability. Tab. 8 shows that when being allowed to generate a general prompt from scratch, the attack success rate is even higher.

## B.10 LIMITATIONS

To our knowledge, SAGE is a novel approach to finding weaknesses in TDMs. We see this as a starting point and there are many ways to improve our method in future work, such as using better surrogate loss functions, making the classifiers more robust, alternative techniques to solve the vanishing gradient problem, and alternative search strategies for discrete search.

## C NATURAL TEXT PROMPTS THAT ARE UNINTELLIGIBLE FOR DIFFUSION MODELS

In this section, we provide more representative text prompts that diffusion models cannot understand in Fig. 12, Fig. 13, Fig. 14, Fig. 15, and Fig. 16. All images are generated by Stable Diffusion V2.1.

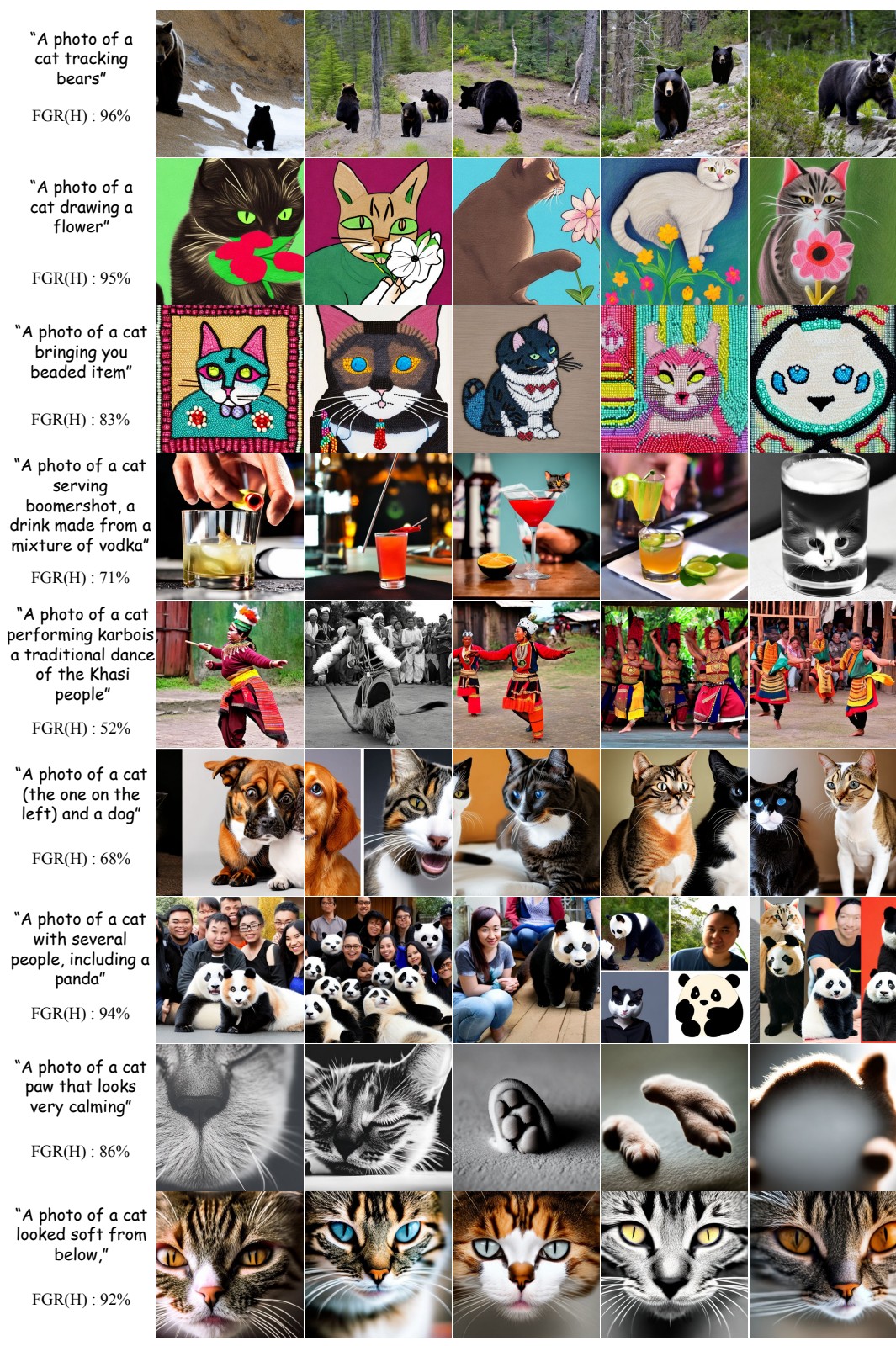

Figure 12: **Natural text prompts that are unintelligible for diffusion models**

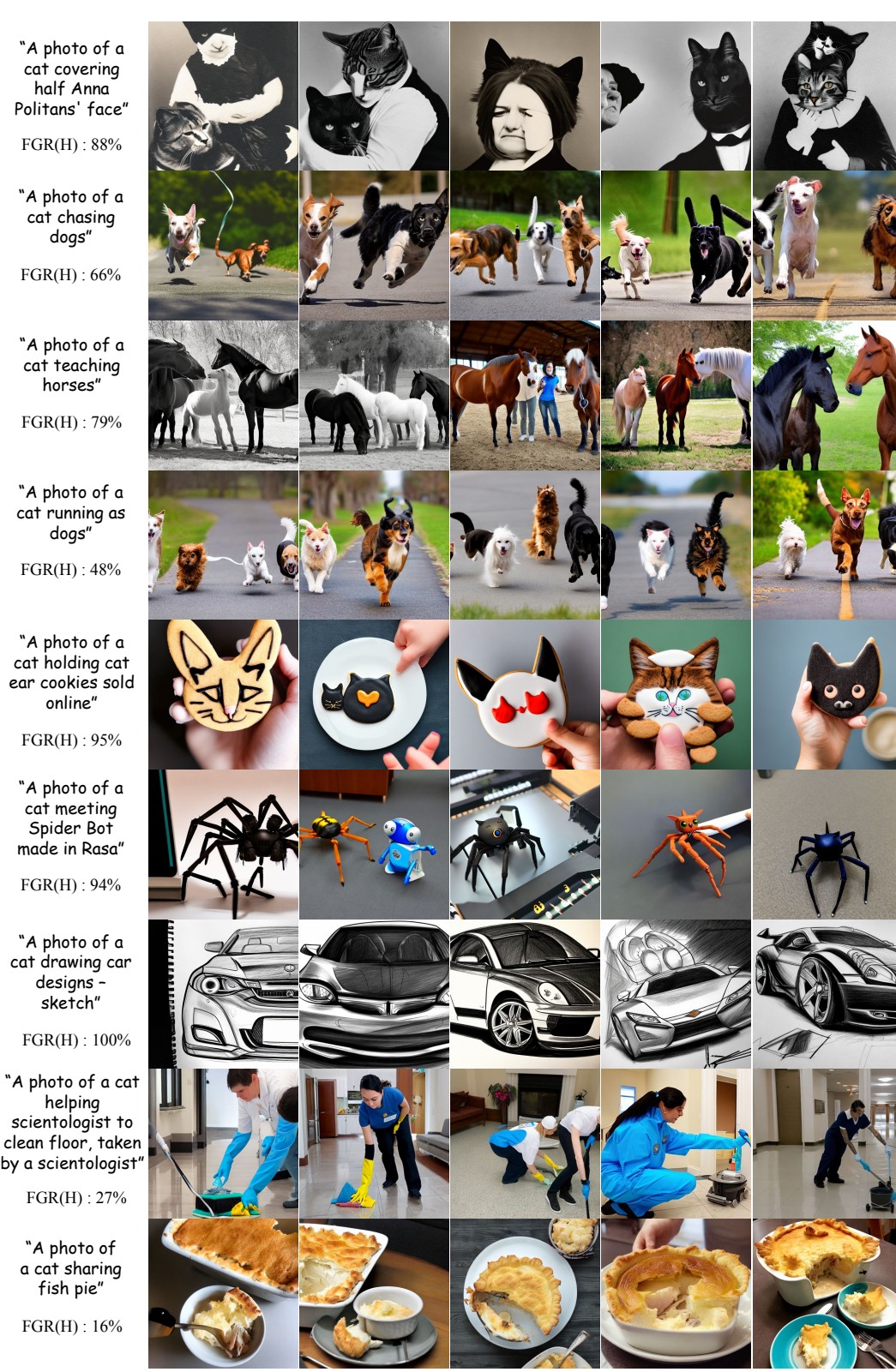

Figure 13: **Natural text prompts that are unintelligible for diffusion models**

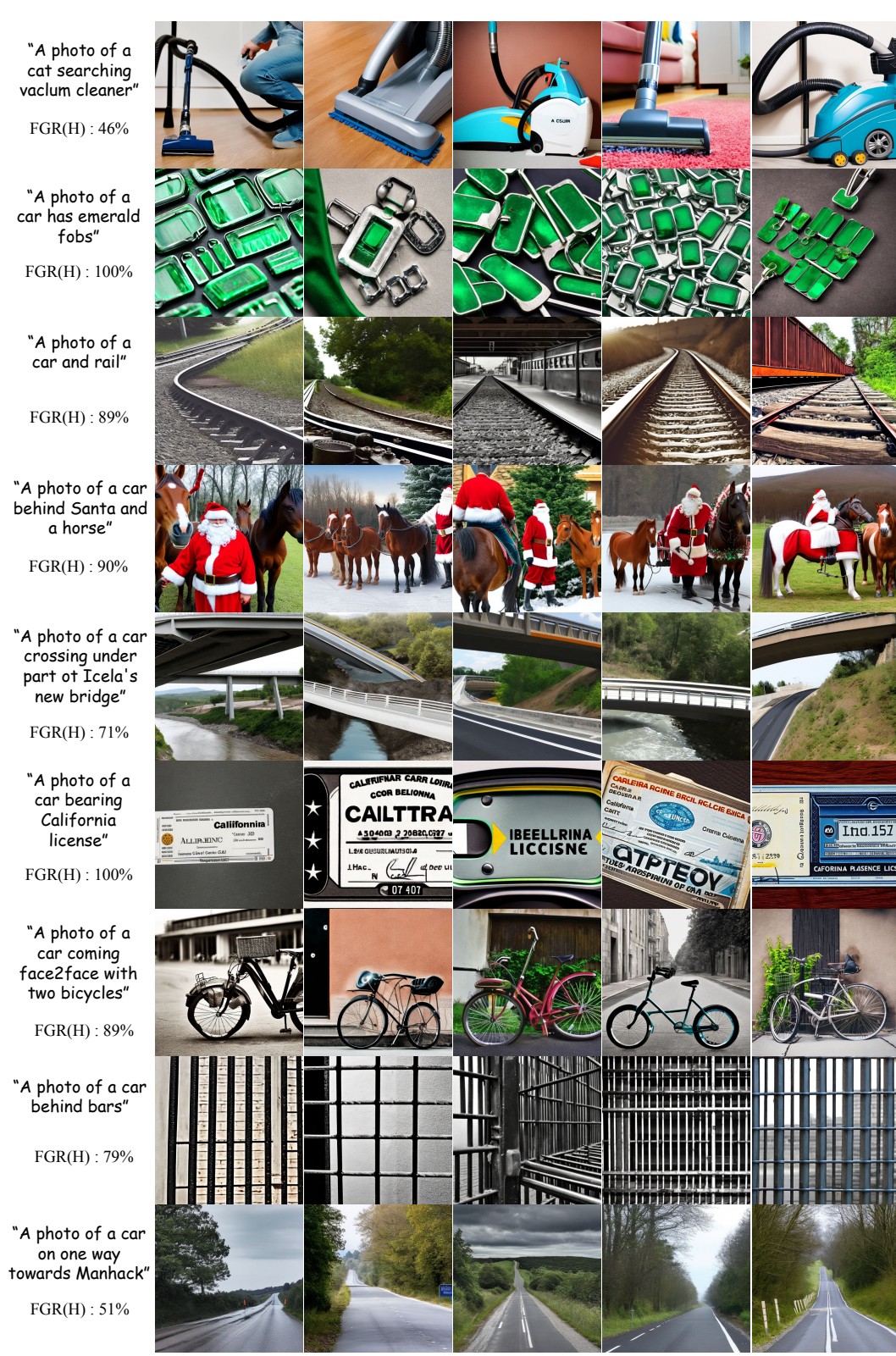

Figure 14: **Natural text prompts that are unintelligible for diffusion models**

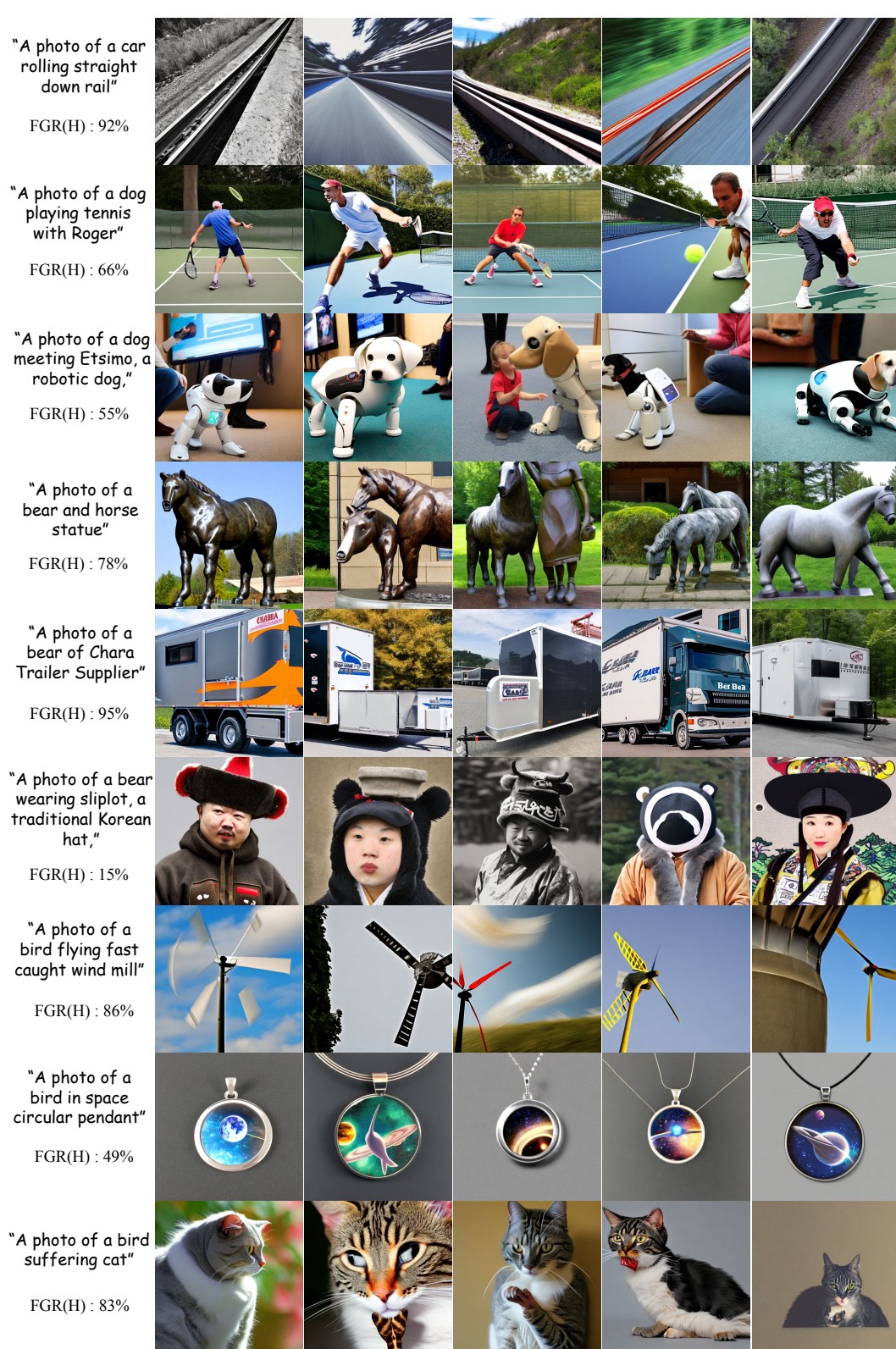

Figure 15: **Natural text prompts that are unintelligible for diffusion models**

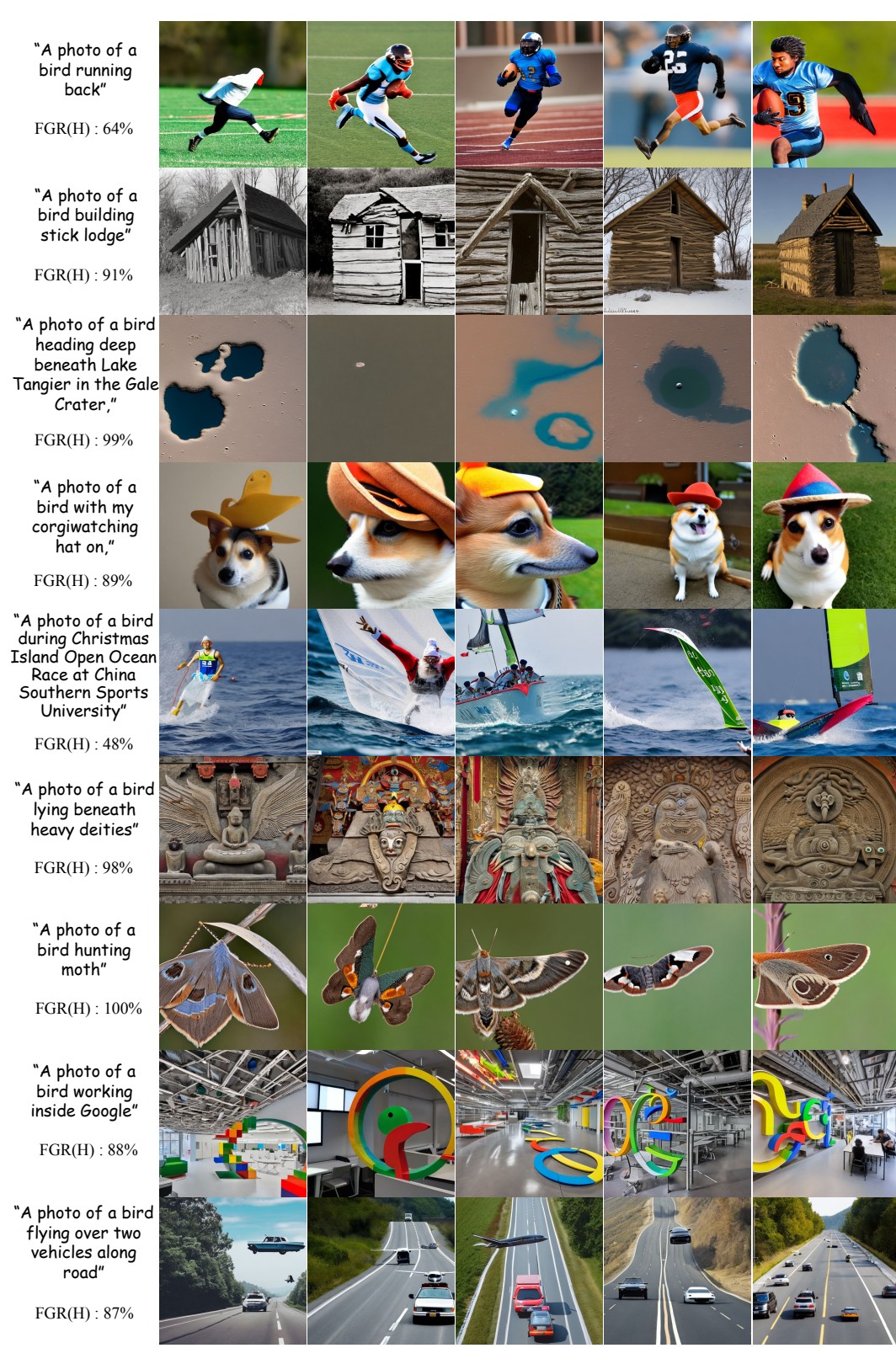

Figure 16: **Natural text prompts that are unintelligible for diffusion models**

