# OpenReview forum: "Discovering Failure Modes of Text-guided Diffusion Models via Adversarial Search"
_ICLR.cc/2024/Conference — ICLR 2024 poster_

### Official Review · Reviewer_jsKc · 2023-10-19

**Soundness:** 2 fair
**Presentation:** 1 poor
**Contribution:** 2 fair
**Rating:** 6
**Confidence:** 4

**Summary:**

In this study, the authors investigate the limitations of Text-guided diffusion models (TDMs) and propose a novel approach called SAGE to systematically explore and understand these failures. TDMs are commonly used for image generation but can exhibit unexpected issues. The study identifies four key failure modes that have not been extensively studied before:
1) TDMs can generate images that fail to accurately represent the semantics of the input text prompts. The authors discuss the causes and potential solutions for this issue.
2) Some regions in the latent space of TDMs lead to distorted images, regardless of the text prompt. This suggests that certain parts of the latent space are not well-structured.
3) Latent samples can produce natural-looking images that are unrelated to the given text prompt, indicating a potential misalignment between the latent and prompt spaces.
4) The addition of a single adversarial token embedding to input prompts can lead to the generation of various specified target objects with minimal impact on CLIP scores, highlighting the fragility of language representations.

Overall, the SAGE method efficiently explores both the discrete language space and the complex latent space, shedding light on these TDM failure modes and offering insights into potential solutions.

**Strengths:**

1. This work is interesting and systemically studying the failure mode of text to image generation model is an important and sometimes overlooked area of research.
2. Comprehensive human study provides important validation on the results.

**Weaknesses:**

1. It's well known that text image generative model can not handle multiple objects especially when keywords are relational and containing novel actions, for instance, “A photo of a cat fighting a fish”. If it's a novel scenario that was not seen in the training set, text to image models often produce only one of the subjects or some blended versions.

2. Most of the proposed failures seem contrived, which are not really issues concerning day to day usage of text-to-image models, especially the ones in latent space, where explicitly optimization/distortion need to be performed on the latent vector, such that it will produce distort the image. If sampled naturally, this event is very unlikely to happen.

3. The arguments are very hand-waving. Not enough evidence/details are provided to support the claims.
    1) "Furthermore, we demonstrate that these failure examples are not isolated spots in the latent space; rather, they exist as connected regions, wherein any sample will generate similar distorted images (see Supp. B.3). It shows that these distorted images can be generated with meaningful probability under Gaussian sampling, and the probability can be computed by the probability density function."
    No calculation has been shown/referred to neither in the main body nor the appendix on the probability. In addition, the paper demonstrated only QQ-plot and statistic of three prompts. It's unclear whether it's a generalized phenomenon.
     2)  "Tab. 2 compares the normalized size of these failure regions and the probability of ..." No details are provided on how the failure regions were calculated/estimated. How exactly the algorithms in PoseExaminer is adopted remains questionable.



One side note but not the main concerns I have on this paper: the presentation lacks structure and appears to be very messy. The authors seem to be ambitious in delivering many things all at once but failed to fufill any of the promises.

**Questions:**

1. Do models hidden behind APIs also suffer the same failure modes?
2. Instead of using the proposed optimization method to "failure mode", which is not of much significant value, why not use it to show the capability of steering/manipulation of the generated images?
3. What happens if prompt engineering was applied, would it impact the way it fails?

---

> ### Author Response · Authors · 2023-11-20
> **Author's Response to Weaknesses Raised by Reviewer jsKc (1/2)**
>
> Based on the reviews, we notice that the main contribution of our paper was not fully appreciated, since the review mostly focuses on the failure patterns we reported.
>
> We want to emphasize that the main contribution is **SAGE, the first automated method to systematically and effectively find failure modes in any TDM.** SAGE comprises three non-trivial types of attacks in the latent space, token space, and human-understandable language space. Achieving these three attacks is challenging, due to the discrete nature of human language space and its intricate patterns, as well as the gradient vanishing problem in the latent space, making the optimization non-differentiable directly. **As appreciated by all other three reviewers, this problem we introduced is novel, meaningful, and important in the research of TDMs. And the proposed method, i.e SAGE,  is effective (6z3X), intelligent (6EFD), and practical (6EFD).** The systematic analysis of the failure modes is considered as our second contribution.
>
> &emsp;
>
> > W1: "Multi-object relation is a well-known problem, especially in novel scenarios"
>
> Firstly, **we propose the first algorithm that systematically finds these prompts. Knowing these problems exist is different from finding these prompts automatically. And the latter is even more important** as it enables us to understand the performance of current methods in existing challenging scenarios. If they still fail, our method will further help us to know when, why, and how such failures occur. It helps us find solutions to circumvent these failures in real world applications, and develop better models in the future. For example, Imagen shows that using LLM can significantly improve this mult-object problem, and they show many better examples, but our experiments in Tab 3 show that multi-object is still a problem for LLM-based models. Then, with a comprehensive analysis as we did in the Appendices B.2, we can further understand more about these failures and how we may improve them. **Therefore, we believe that automatically and systemically detecting these errors, no matter for a newly developed method or a widely used product, is very important and helpful.**
>
> **Secondly, multi-object relation is just one of the ten typical failures of incomprehensible language prompts we reported in Fig 4.** As we discussed in Sec 5.1 and Sec B2 (a) , the failure of “cat fighting a fish” **is not due to multiple objects, but due to the actions itself.** We also give an example “a bird running back”, which only has one object, but still fails due to the action. In fact, **at least 7 types of failures in Fig 4 (a,b,c,e,h,i,j) contain prompts that only involve one single object** (Please see Sec B2).
>
> Finally, we want to argue that **the ability to generate such unseen/novel scenarios is in fact one of the most important features of TDMs.** Many showcases on the official websites of Imagen and Parti involve uncommon prompts and multiple objects (e.g., “A brain riding a rocket ship heading towards the moon”, etc ). We think **any scenarios that can be easily imagined by humans should be able to be generated by a TDM.** In addition, many prompts we reported can be found in real world. For example, “a cat fighting a fish” can be envisioned as your cat trying to catch a goldfish in your fish tank.
>
> &emsp;
>
> > W2: Failures in latent space seems contrived and are not really issues in day to day usage.
>
> Firstly, as we discuss in Sec 5.3and Tab 2, **the failure in latent space can be sampled in day-to-day usage, and is related to model stability** (i.e., the probability of unrelated images being generated under random sampling). The method we presented provides an efficient way to reveal these problems in advance. And it enables us to evaluate which object categories are not sufficiently well represented. For example in Table 2, for the 5 categories found by SAGE, **9.2% of the randomly generated images are irrelevant to the inputs**, which is clearly important information.
>
> In addition, the existence of these latent variables indicates that the latent space is not well-structured, which is important and provides insights for further improvement of diffusion models. As shown in other papers [b,c], understanding the latent space is critical for downstream tasks.  Our method provides new insights into the latent space, and opens a new solution to study it.
>
> Finally, we want to mention again that the types of failures discovered in latent space are just a small part of our second contribution. The main contribution is SAGE, the first automated method to systematically find failure modes of any TDMs. The second contribution is the failure modes we discovered, which includes all ten types of failures in prompt space and most of them are unexplored previously. Failure in latent space is just one small part of our second contribution.
>
> [b] Blattmann et al. "Align your latents: ..." CVPR 2023.
>
> [c] Xia et al. "Gan inversion: A survey." TPAMI 2022

---

> ### Author Response · Authors · 2023-11-20
> **Author's Response to Weaknesses Raised by Reviewer jsKc (2/2)**
>
> > W3: The arguments are very hand-waving.
> >> W3.1: "No calculation has been shown/referred to neither in the main body nor the appendix on the probability. In addition, the paper demonstrated only QQ-plot and statistic of three prompts. It's unclear whether it's a generalized phenomenon."
>
> Regarding the **calculation of the probability**, what we want to emphasize is that since the failure samples exist as a connected region (e.g. $[a,b]$), then **the sample rate is not zero and is theoretically computable through the probability density function:**
> $$\text{Pr}[a\leq X \leq b] = \int_a^bf_X(x)dx$$
> It implies that these failure cases are meaningful. However, computing the exact probability is impossible since it is an extremely high-dimension problem and the exact range of each dimension is hard to determine. We will revise this sentence into :"*These failure examples can be generated with non-zero probability under Gaussian sampling according to the probability density function.*”
>
> &emsp;
>
> Regarding whether it is a **generalized phenomenon. It is indeed a generalized phenomenon.** For **ALL** results, we compute their statistics to evaluate if they are outliers from $\mathcal{N}(0,I)$, and report the overall **Non-Gaussian Rate** in Table 1.  **75% percent are not outliers, showing that it is actually a generalized phenomenon.**
>
> &emsp;
>
> >>W3.2: "No details are provided on how the failure regions were calculated/estimated"
>
> PoseExaminer provides an intuitive way to compute the final boundary in a high-dimension system under certain constraints, in which they gradually expand the upper and lower boundary of each dimension. We follow their method, and starting from the latent variable we find, we gradually expand the boundary until it becomes an outlier or no longer causes the model to fail. We add a detailed explanation in Appendices B.3 of the revision.
>
> &emsp;
>
> > W4: "One side note but not the main concerns I have on this paper: the presentation lacks structure and appears to be very messy. The authors seem to be ambitious in delivering many things all at once but failed to fufill any of the promises."
>
> Yes, we indeed try to thoroughly discuss all failures we find, delving into the underlying causes and potential solutions. Our manuscript underwent multiple revisions prior to submission. And we believe reviewers 5bJZ, 6z3X, and 6EFD agree that our paper is well-structured and easy to follow (as indicated in their comments and the Presentation score).  We would greatly appreciate any additional detailed suggestions to further refine this paper. Thank you very much!

---

> ### Author Response · Authors · 2023-11-20
> **Author's Response to Questions Raised by Reviewer jsKc**
>
> We answer the questions raised by Reviewer jsKc below.
>
> &emsp;
>
> >Q1: "Do models hidden behind APIs also suffer the same failure modes?"
>
> Yes. We have experimented with StableDiffusion-XL online, DALL-E, Firefly from Adobe, and Midjourney. They also suffer from many similar failures. For example, StableDiffusion-XL struggles with scenarios like "cat tracking bear," while Midjourney faces difficulties with the concept of a "beaded item."
>
>
> In addition, we want to emphasize that our method is not intended to “attack” an already deployed system. **Instead, its purpose is to help researchers and developers to understand the important failure modes of the method they are working on,** helping develop better image diffusion models. Just like the role of a standard benchmark for discriminative tasks like classification or detection, understanding the failure modes of a machine learning system is generally important.
>
> &emsp;
>
> >Q2: "Instead of using the proposed optimization method to failure mode, which is not of much significant value, why not use it to show the capability of steering/manipulation of the generated images?"
>
> This is a very good point. We think our method can be directly used to manipulate/generate more accurate and realistic images, by optimizing the diffusion in an opposite direction instead of making them fail.
>
> However, **as acknowledged by the other three reviewers, we still believe that understanding the failure modes of TDMs is a meaningful and important problem in the research of TDMs.** It provides insights into how to further improve current TDMs, and provides a benchmark to evaluate any future TDM, which could be a long-term and important benefit for the development of TDMs. However, we do agree that using our method to improve the generated image is an interesting idea, and we will explore it in the future. Thank you!
>
> >Q3: "What happens if prompt engineering was applied, would it impact the way it fails?"
>
> This is a good question. It mainly depends on the type of failure modes. For example, prompt engineering can help alleviate the failure caused by certain actions, by giving a more detailed description of the action. For example: ‘A photo of a cat fighting fish’ generated wrong images in SD2.1, but ‘A photo of a cat and a fish, the cat is trying to catch the fish’ gives better results. However, finding these prompts can be very hard.
>
> In addition, prompt engineering can not address many failures caused by the failure of the diffusion module itself, such as certain deformation of the object required by the prompt (see Appendices B.2). For these questions, a better and more robust diffusion model is needed.

---

> > ### Comment · Reviewer_jsKc · 2023-11-22
> > **Thank you for the responses**
> >
> > I have read the responses carefully and understand the motivation a bit more. Although I still can not fully appreciate the said contributions, I am willing to raise the rating to weak accept.

---

> > > ### Author Response · Authors · 2023-11-23
> > > **Thank you**
> > >
> > > Thank you for your response! We really appreciate your careful inspection and invaluable suggestions.

---

> ### Author Response · Authors · 2023-11-22
>
> Dear Reviewer jsKc,
>
> Thanks again for your valuable feedback and suggestions. We are curious whether the feedback we provided has effectively addressed your concerns. Feel free to add new comments if you have any further questions. We are more than happy to continue discussions and will do our best to provide thorough responses.
>
> Best regards,
>
> Paper 1931 Authors

---

### Official Review · Reviewer_6EFD · 2023-10-30

**Soundness:** 4 excellent
**Presentation:** 4 excellent
**Contribution:** 4 excellent
**Rating:** 10
**Confidence:** 5

**Summary:**

Text-guided diffusion models (TDMs) are prone to unexpected failures, such as generating incorrect images from natural-looking text prompts or producing inconsistent images from the same text with different latent variable samples. To address this, the study introduces SAGE, an adversarial search method that explores TDMs' prompt and latent spaces to identify failures, using image classifiers for guidance and human verification for accuracy. The investigation reveals four main failure modes, highlighting issues with semantic capture, latent space structure, prompt-latent misalignment, and the fragility of language representations, suggesting avenues for future improvement.

**Strengths:**

- Proposed a smart adversarial search algorithm to identify failures in TDMs, finding “reasonable” tokens, texts, and latent codes to input into the TDMs, leading to failure generations.

- Here, “reasonable” latent code implies that we draw the code close to N(0,I), and the text should be human-readable. This proposed adversarial attack is not only fair to the TDMs but also practical for users.

- The algorithm demonstrates an intelligent method (using residual connections) to avoid backpropagating through the entire extensive diffusion model, thereby enhancing computational efficiency.

- The text is well-composed, accompanied by a self-explanatory figure delineating the overall pipeline.

- The system is fully automated and reinforced by human evaluation.

**Weaknesses:**

- Given that the pipeline heavily relies on the robust classifier, I'm curious if this means the SAGE system will primarily detect the blatantly poor cases, while struggling to identify the more subtle, not-so-good ones.

- While SAGE employs an ensemble of classifiers, it still seems akin to a handful of neural classifiers trained on the same dataset. True, one might outperform another, but I'm inclined to think that these classifiers would exhibit similar decision boundaries.

**Questions:**

1. Q1:
- I'm somewhat puzzled by the SSR(A) in Table 1. It seems slightly biased in favor of the SAGE method, considering it's directly optimized based on this metric.

2. Humans and automatic systems have different perceptions.

- Q2.1: Would you attribute the reason SSR(H) isn't a full 100% to SAGE generating some false positives, or is it more about the perceptual differences among the observers?

- Q2.2: As I mentioned earlier, my concern is that SAGE may only identify blatantly incorrect instances. From what you've observed, were there moments where you thought, "That's clearly a failure, but SAGE didn't recognize it in the samples"? If such instances occurred, could you provide some insights into why that might be?

Overall this is an excellent work, I can’t wait to try your demo.

---

> ### Author Response · Authors · 2023-11-20
> **Author's Response to Concerns Raised by Reviewer 6EFD**
>
> Thank you so much! We are thrilled to hear that you enjoyed my work. It is truly encouraging.
>
> &emsp;
>
> >W1: "If SAGE primarily detects the blatantly poor cases"
>
> Yes, it is correct. All the failures we detect are obvious failures, where the key object is not observed in the generated image. That’s why in Table 1, human evaluation often reports more failures than automated evaluation. There are several failure cases that cannot be detected by automated evaluation, such as generating incorrect objects (e.g. a cat with two heads), incorrect actions, incorrect numbers, and incorrect features (e.g. ‘ white cat’ generating a yellow cat). These failures pose challenges for current algorithms, requiring better and more robust discriminative models—many of which do not exist. In fact, the evaluation of generative models is still an open question, with current metrics like FID score and CLIP score proving limited and less accurate. It is worth noting that SAGE is general and the evaluator can be changed easily. We expect that future research will find even better discriminative models.
>
> In addition, we believe this actually shows the effectiveness of our model. Despite focusing on these obvious failure cases, we still achieve a high search success rate.
>
> &emsp;
>
> >W2: "These classifiers would exhibit similar decision boundaries."
>
> That is another reason to add an edge-based classifier. While it has a relatively lower classification accuracy compared to a standard ViT, **it has a rather different decision boundary.** Standard ViTs often exhibit bias toward object textures over shapes [a], while the edge-based classifier focuses solely on object shapes. It greatly reduces the false positive rate and helps us find failures genuinely generated by the diffusion model. However, we agree that integrating more discriminative models with different decision boundaries would further improve our algorithm, and even enable us to find more subtle cases.
>
> [a] Geirhos et al. "ImageNet-trained CNNs are biased towards texture..." ICLR. 2018.
>
> &emsp;
>
> >Q1: "SSR(A) seems slightly biased in favor of SAGE"
>
> In fact, both the baseline and SAGE use this metric as the optimization goal, so we think it is a fair comparison. We use it because it is the only robust metric we currently have that gives less false positives. As demonstrated in Figure 3 and the reported number on it, the ensemble of robust classifiers still gives a false positive rate of 67.5%, while our classifier achieves a significantly lower false positive rate of only 22.5%.
>
> &emsp;
>
> >Q2: "Humans and automatic systems have different perceptions."
> >>Q2.1: "The reason SSR(H) isn't a full 100%"
>
> In short, the diffusion model generates many ambiguous images with strong disturbance that fool the discriminative model, resulting in a 100% SSR(A). However, during the human evaluation, we categorize all ambiguous images as correct cases, which cause SSR(H) not a full 100%.
>
> Specifically, despite our efforts to build a robust classifier system, the most robust discriminators remain vulnerable. When optimizing the latent variables, the diffusion model can generate challenging cases with strong disturbance for the discriminators. However, many of these challenging cases are ambiguous even for humans. In general, we set a very lenient threshold for human evaluators: Since the input prompt is ‘A photo of a [class]’, we consider any image with key features of the target objects as a correct case. For example, in Figure 3, we consider the left image as a cat while the right one is not.
>
> >>Q2.2: The reason why SAGE didn't recognize a failure observed by human:
>
> Yes, we have indeed observed many of these cases. We summarize several of them here:
>
> 1. A prompt typically contains several features beyond the object itself, such as colors, relations, actions, etc. However, we do not have a good discriminative model to detect all these features. Understanding all information from an image involves many open problems in computer vision. An example of this is in Figure 4 (b), what we want is a real cat near a sketch of a flower. However, the generative model often gives sketches of both cat and flower, and the discriminative mode can not tell the difference. The answer in W1 gives more examples.
>
> 2. Current discriminative models usually detect whether there is any feature of the target object in the image. Therefore, some dominant features alone can be identified as the target object with a very high confidence score. This is completely different from human vision. As exemplified in Figure 4 (h), many generated images contain a human's hand/foot with cat skin, yet these images are still classified as a cat. Similar problems can also be found in Figure 5 (b) and (c), where a dalmatian and a warplane are detected.
>
> 3. Current discriminative models do not assess the reasonableness of the object. A cat with two heads or a person with four legs will still be recognized by the discriminative model.

---

> ### Author Response · Authors · 2023-11-22
>
> Dear Reviewer 6EFD,
>
> Thank you once again for your positive feedback and valuable suggestions! If you require any additional information or clarification from us, please don't hesitate to reach out.
>
> Best regards,
>
> Paper 1931 Authors

---

### Official Review · Reviewer_6z3X · 2023-11-01

**Soundness:** 3 good
**Presentation:** 3 good
**Contribution:** 2 fair
**Rating:** 6
**Confidence:** 3

**Summary:**

This paper presents an automatic way of detecting failure text prompts in text-guided diffusion models (TDMs). A robust image classifier based surrogate loss is proposed to detect accurate failure cases due to diffusion models. In addition, to deal with vanishing gradient issue, the authors apply approximate gradients to back-propagate, via residual connection. Adversarial based approach helps to identify the natural text prompts (non-outlier) that cause the failure cases. The experimentation results show the model is efficient in finding the failure cases (SSR), and evaluated via human annotation.

**Strengths:**

I think the task the paper is presenting is novel and is important in the research of TDMs. The authors managed to present thorough experimentation and analysis on evaluating the proposed model performance. The presented samples do show the effectiveness of model in identifying the true failure cases from natural text prompts.

**Weaknesses:**

1. Overall, while the paper is one of the first to target such a problem in finding actual failure in TDMs, the problem itself is rather similar to some of the previously well studied tasks, with adversarial-based approaches. The discriminator here is to identify and remove the irrelevant feature in the latent space, so that it becomes task oriented. Such an approach has been used in tasks such as fair classification.
2. Though various evaluation being conducted, it is hard to measure the effectiveness of each proposed component in the model. To better connect the intuition of each contribution of the work and the actual analysis/effectiveness, it would be great to have some ablation study.

**Questions:**

1. As many findings are presented in the paper, what would be some potential solution to handle these failure cases?

---

> ### Author Response · Authors · 2023-11-20
> **Author's Response to Concerns**
>
> We thank Reviewer 6z3X for the review, and we address the concerns below.
>
> &emsp;
>
> > W1: "While the paper is one of the first to target such a problem in finding actual failure in TDMs, the problem itself is rather similar to some of the previously well studied tasks."
>
> While the high-level concept is similar to other tasks, like adversarial machine learning in image classification, we believe it does not diminish the novelty or contribution of our paper. As the reviewer already appreciated, we define and study a novel, important, and meaningful task in the context of text-guided diffusion models (TDMs), namely discovering the failure modes in TDMs automatically and systematically. We provide an effective and comprehensive solution to solve this task, and reveal several previously unexplored but important failure modes that exist across all widely used TDMs.  Moreover, finding these failure modes in TDMs automatically is very challenging, leading previous researchers to rely on manual sampling:
>
> 1. The human language space is discrete, making it hard to optimize. It also follows intricate patterns, which requires complex constraints to ensure that the generated prompt remains natural and comprehensible for humans. To solve this problem, we use LLM as the language prior, and propose a gradient-guided optimization to efficiently search over the discrete language space.
>
> 2. Unlike GANs or classification models, diffusion models are significantly more complex, often involving hundreds of sampling steps during inference.  This complexity makes it impossible to directly back-propagate the gradient from the output to the latent space. To solve this problem, we add residual connection in the diffusion step and use the approximate gradients for efficient optimization.
>
> 3. We do not have a reliable metric for image generation or a robust discriminator to detect failures. Currently, classifiers are sensitive to noise and biased towards texture. A simple ensemble of classifiers gives us a high rate of false positives (i.e., most failures are due to the classifier errors instead of the TDMs). To solve this problem, we train an edge-based classifier that has different decision boundaries.
>
> In the experiment section, we demonstrate the effectiveness of all these methodological components through various ablation studies (Please also see the answer for W2)
>
>
> In conclusion, as you already appreciated, the task we introduce is novel and important, and is very challenging. Our proposed method is also novel, non-trivial, and effective, despite sharing a very high-level concept with other adversarial-based approaches.
>
> &emsp;
>
> > W2: "It would be great to have some ablation study."
>
> Yes, we agree that the ablation study is important. **We provide several ablation studies in the original paper**. However, due to space constraints, **most of the ablations are included in the appendices**. We list all ablations in the paper and the appendices as follows:
>
> 1. The baseline in Table 1, under the section “Human-understandable text prompt space”, uses greedy search with LLaMA instead of the proposed gradient-guided search with LLaMA. The observed improvement shows *the effectiveness of gradient-guided optimization.*
> 2. The baseline in Table 1, under the section “Latent space”, uses a modified projected gradient descent instead of the proposed searching policy in latent space. We apply several engineering optimizations to fit the entire system with a standard PGD in an A100. The improvement shows *the effectiveness of using approximate gradients with residual connection.*
> 3. Figure 3 and the reported search success rate on the figure compare the results with and without edge ViT, showing *the importance of the edge-based classifier.*
> 4. Table 5 and Figure 10 compare the generated image with and without residual connection, showing that the residual connection does not affect the generated image. Table 6 compares the search success rate when adding residual connections to different denoising steps. These two ablation experiments *demonstrate that residual connection enables efficient searching of failure cases in the latent space without affecting the image generation.*
> 5. Table 7 compares *the different choices of large language models*.
> 6. Table 8 compares *the use of prompt template* (i.e., “A photo of a” )
>
> We have tried our best to provide elaborate ablation studies. We would greatly appreciate more specific suggestions regarding additional ablations that could enhance our analysis.

---

> ### Author Response · Authors · 2023-11-20
> **Author's Response to Questions**
>
> We address the question raised by Reviewer 6z3X below.
>
> &emsp;
>
> > Q: "What would be some potential solution to handle these failure cases?"
>
> We discuss in detail the **underlying causes and several potential solutions of each failure mode in Appendices B.2.** For example, in the first failure reported in Figure 4 (i.e., failure due to specific actions), we have identified three potential causes: (1) the incorrect pairing of specific verbs with nouns (e.g., “a cat fighting fish” consistently interpreted as a type of fish in SDV2.1, but not in SDV1.5), (2) the bias of certain objects with actions (e.g., “a bird running back” usually results in an image of a man running, but with feathers on his arms), and (3) the deformation of objects related to the actions (e.g., “a cat chasing dogs” usually gives an image of an unidentifiable object chasing a dog ).
>
> Considering the underlying causes, we think that problem 1 can be alleviated by better prompt engineering. Additionally, using LLM instead of CLIP, or adopting a better training strategy for CLIP to disentangle object-level information with attributes, may alleviate both problem 1 and 2.
>
> However, we want to emphasize that finding a good solution is difficult. As shown in Table 3, although LLM helps alleviate some problems, it doesn't completely solve them; rather, it merely reduces the failure rate.
>
> Please take a look at Appendices B2 for more discussions.

---

> ### Author Response · Authors · 2023-11-22
>
> Dear Reviewer 6z3X,
>
> Thanks again for your valuable feedback and suggestions. We are curious whether the feedback we provided has effectively addressed your concerns. Feel free to add new comments if you have any further questions. We are more than happy to continue discussions and will do our best to provide thorough responses.
>
> Best regards,
>
> Paper 1931 Authors

---

### Official Review · Reviewer_5bJZ · 2023-11-01

**Soundness:** 3 good
**Presentation:** 3 good
**Contribution:** 3 good
**Rating:** 6
**Confidence:** 4

**Summary:**

This paper proposed a method called SAGE to search natural and human-understandable texts that text-guided diffusion models (TDMs) cannot generate images correctly for the first time. This method explores the discrete prompt space and the high-dimensional latent space to discover undesirable behaviors automatically. This method utilizes algorithms of adversarial attack and image classifiers as surrogate loss functions. To generate natural prompts, the authors use large language models (LLMs) like LLaMA to search for suitable prompts. The authors conduct experiments on several metrics to demonstrate the effectiveness. The authors also conclude 4 different failure types of TDMs by analyzing the results of failure examples.

**Strengths:**

1.	The topic is meaningful for researchers to understand the failures of diffusion models. Natural prompts are closer to real-world scenarios and are more beneficial for improving the robustness of TDMs.

2.	This work further analyzes the deeper causes and possible solutions through the structure of TDMs and the corresponding language features. These discussions appear to be both comprehensive and effective.

3.	The paper is well-structured and easy to follow. The procedure is demonstrated well in Figure 2 and its description. The author provided detailed descriptions of the method and experiments.

**Weaknesses:**

1.	The authors do not demonstrate their method in pseudo-code. The code is not included in the supplement material either. Would the authors demonstrate their method in pseudo-code?

2.	Time cost is not shown in the paper.
It seems time-consuming to run a LLaMA and an ensemble of classifiers simultaneously in the attack. How much GPU memory and how long does it take to find an example?

3.	The detail of human evaluation is missing.
The paper doesn't demonstrate differences in the ratings of different human evaluators. How does the author handle rating differences and assess the accuracy of human evaluators?

4.	The metric needs to be clarified.
Could the authors further explain why the Non-Gaussian Rate (NGR) is reported? What is the purpose of this metric in the experiments?

**Questions:**

1.	How much GPU memory and how long does it take to find an example?

2.	Could the authors further explain why the Non-Gaussian Rate (NGR) is reported? What is the purpose of this metric in the experiments?

3.	How does the author handle rating differences and assess the accuracy of human evaluators? Could the authors further explain the human evaluation process?

---

> ### Author Response · Authors · 2023-11-20
> **Author's Response to Concerns Raised by Reviewer 5bJZ**
>
> We thank Reviewer 5bJZ for the review, and we address the concerns below.
>
> &emsp;
>
> >W1: Pseudo-code
>
> Thank you for the valuable suggestion. We will provide the pseudo-code in Appendices A.2 of the revision. We plan to submit the revision on Monday, Nov 20. Additionally, the code will be released upon the publication of this paper.
>
> &emsp;
>
> >W2 (Q1): Time and memory cost
>
> For finding natural prompts, it takes an average of 51.26 minutes on one A100 GPU. LLaMA itself requires about 24GB of memory, the SD V2.1 we test requires around 10GB. The ensemble of classifiers takes about 12 GB, and the entire system requires approximately 46 GB of memory.
>
> &emsp;
>
> > W3 (Q3): "The detail of human evaluation is missing"
>
> We mentioned the human evaluation details in Appendices B.1. In summary, 3 human evaluators are assigned to each image, and the final score for each image is the average of their ratings. During the rebuttal period, we computed the variance of the score obtained for each image. We report the results in the following table. Notably, the ratings exhibit good consistency for the majority of the images. We will also include this table in the revision.
>
> | Variance        | <=0.222 | 0.222-1 | >=1  |
> |-----------------|--------------|--------------|-----------|
> | Proportion     | 76.2%   | 23.4%   | 0.4% |
>
> &emsp;
>
> > W4 (Q2): "why the Non-Gaussian Rate (NGR) is reported?"
>
> We report the Non-Gaussian Rate as it measures the validity of failure cases in latent space. Typically diffusion models assume the latent variable follows an $\mathcal{N}(0,I)$ distribution. Therefore, we want the failure samples to follow this distribution, i.e. to have a high likelihood under the  $\mathcal{N}(0,I)$ distribution, which indicates that these samples are not outliers.

---

> > ### Comment · Reviewer_5bJZ · 2023-11-23
> >
> > Thank you for the response. My questions have been addressed.

---

> ### Author Response · Authors · 2023-11-22
>
> Dear Reviewer 5bJZ,
>
> Thanks again for your valuable feedback and suggestions. We are curious whether the feedback we provided has effectively addressed your concerns. Feel free to add new comments if you have any further questions. We are more than happy to continue discussions and will do our best to provide thorough responses.
>
> Best regards,
>
> Paper 1931 Authors

---

### Author Response · Authors · 2023-11-20
**Author Feedback for All Reviewers**

&emsp;


Dear Reviewers and ACs:


&emsp;


Thank you so much for your time and efforts in assessing our paper. We hope our rebuttal has addressed your concerns.  If you still have other concerns, we are more than happy to continue discussions and will do our best to provide thorough responses. Thanks again for helping improve our paper.


&emsp;


**Rebuttal revision**:  We are truly grateful for the constructive comments and insightful reviews that have greatly enhanced our initial manuscript. We have carefully incorporated all the suggestions, making major changes to the previous draft. The key modifications are highlighted in blue throughout the revision.


Specifically, we have made the following key modifications:


1. Added the Pseudo-code in Sec. A.2 and Algorithm 1 (Reviewer 5bJZ)
2. Provided additional details for human evaluation in Sec. B.1 and Table R-1 (Reviewer 5bJZ)
3. Updated details regarding the calculation of probability in Sec. 5.2 (Reviewer jsKc)
4. Provided additional details on using PoseExaminer to identify failure regions in Sec. B.3 (Reviewer jsKc)

&emsp;

Once again, we greatly appreciate your time and efforts.

&emsp;

Best regards,

Paper 1931 Authors

---

### Meta-Review · Area_Chair_xH37 · 2023-12-06

**Metareview:**

This paper presents SAGE,  an automatic way of identifying failure modes in Text-guided diffusion models (TDMs). The paper explores the discrete text prompt space and the latent space of initial Gaussian noise and finds regions in both spaces that are not outliers but can lead to undesirable behaviors and failure generation cases. Experiments verify the effectiveness of the proposed method, identify common failure modes in TDMs, and suggest future improvements. All reviewers recognize the contribution of this paper in identifying common failure modes for a better understanding of TDMs. Reviewers also agree the evaluations are comprehensive and support the claims. Additionally, reviewers 5bJZ and 6EFD commend the paper for its clarity and the illustrative quality of its figures. During rebuttal, reviewers raised questions or concerns about the practical use cases, runtime costs, evaluation details, and ablation study of the paper. The authors provided detailed responses to the questions, and two reviewers acknowledged that part of their concerns were addressed.

**Justification For Why Not Higher Score:**

The paper did not receive a higher score due to concerns about the lack of detailed evidence and sufficient support for its claims. Additionally, while the paper is acknowledged as one of the first to identify failure modes in TDMs, the problem it tackles is viewed as similar to previously studied tasks involving adversarial approaches.

**Justification For Why Not Lower Score:**

The paper was not assigned a lower score due to its contributions in identifying failure modes in TDMs. Reviewers unanimously recognized the value of this research in enhancing the understanding of TDMs. The comprehensive evaluations, along with the clarity and illustrative quality of its figures, were particularly commended. Additionally, during the rebuttal phase, the authors provided detailed responses to the questions and concerns raised, such as practical use cases, runtime costs, evaluation details, and ablation study, which addressed part of the reviewers' concerns.

---

### Decision · Program_Chairs · 2024-01-16

Accept (poster)